



# Analysis of groundwater flow and stream depletion in the L-shaped fluvial aquifer

Chao-Chih Lin[1], Ya-Chi Chang[2] and Hund-Der Yeh[1]

[1]Institute of Environmental Engineering, National Chiao Tung University, Hsinchu, Taiwan
[2]Taiwan Typhoon and Flood Research Institute, National Applied Research Laboratories, Taipei, Taiwan

*Correspondence to*: Hund-Der Yeh (hdyeh@mail.nctu.edu.tw)

**Abstract.** Understanding the head distribution in aquifers is crucial for the evaluation of groundwater resources. This article develops an analytical model for describing flow induced by pumping in an L-shaped fluvial aquifer bounded by impermeable bedrocks and two nearly fully penetrating streams. A similar scenario for numerical studies was reported in Kihm et al. (2007).
The water level of the streams is assumed to be linearly varying with distance. The aquifer is divided into two sub-regions and the continuity conditions of hydraulic head and flux are imposed at the interface of the sub-regions. The steady-state solution describing the head distribution for the model without pumping is first developed by the method of separation of variables. The transient solution for the head distribution induced by pumping is then derived based on the steady-state solution as initial condition and the methods of finite Fourier transform and Laplace transform. Moreover, the solution for stream depletion rate
(*SDR*) from each of the two streams is also developed based on the head solution and Darcy's law. Both head and *SDR* solutions in real time domain are obtained by a numerical inversion scheme called the Stehfest algorithm. The software MODFLOW-2005 is chosen to check the accuracy of the head solution for the L-shaped aquifer. The steady-state and transient head distributions within the L-shaped aquifer predicted by the present solution are compared with the numerical simulations and measurement data presented in Kihm et al. (2007). The *SDR* solution is employed to demonstrate its use as a design tool in
determining well location for required amounts of *SDR* from nearby streams under a specific aquifer pumping rate.

## 1 Introduction

Groundwater is an important water resource for agricultural, municipal and industrial uses. The planning and management of water resources through the investigation of the groundwater flow is one of the major tasks for practicing engineers. The aquifer type and shape are important factors influencing the groundwater flow. Many studies have been devoted to the
development of analytical models for describing flow in finite aquifers with a rectangular boundary (e.g., Chan et al., 1976; Chan et al., 1977; Daly and Morel-Seytoux, 1981; Latinopoulos, 1982; Corapcioglu et al., 1983; Latinopoulos, 1984, 1985; Lu et al., 2015), a wedge-shaped boundary (Chan et al., 1978; Falade, 1982; Holzbecher, 2005; Yeh et al., 2008; Chen et al., 2009; Samani and Zarei-Doudeji, 2012; Samani and Sedghi, 2015), or a triangle boundary (Asadi-Aghbolaghi et al., 2010). At



present, few solutions are reported to handle the head fluctuation problems for tidal aquifers with re-entrant angle (L-shaped) boundaries (e.g., Sun, 1997; Li and Jiao, 2002), but none of them are to deal with the pumping or stream depletion problems. Many studies focused on the development of numerical approaches for evaluating the groundwater flow in an aquifer with irregular domain and various types of boundary conditions. The rapid increase of the computing power of PC enables the

numerical models to handle the groundwater flow problems with complicated geometric shapes and/or heterogeneous aquifer properties. We therefore adopt the software MODFLOW-2005 to assess the accuracy of the predictions by the present solution. Numerical methods such as finite element methods (FEMs) and finite difference methods (FDMs) are very commonly used in engineering simulations or analyses. For the application of FEMs, Taigbenu (2003) solved the transient flow problems based on the Green element method for multi-aquifer systems with arbitrarily geometries. Kihm et al. (2007) used a general

multidimensional hydrogeomechanical Galerkin FEM to analyze three-dimensional (3D) problems of saturated-unsaturated flow and land displacement induced by pumping in a fluvial aquifer in Yongpoong 2 Agriculture District, Gyeonggi-Do, Korea. The domain of the aquifer is in L shape and bounded by streams and impermeable bedrock. They performed FEM simulations for steady-state spatial distributions of hydraulic head before aquifer pumping and then for the distributions of hydraulic head and land displacement vector after one-year pumping. Both simulation results were compared and validated with the field

measurements of hydraulic head and vertical displacement in transient case.

The FDMs have been widely utilized in the groundwater problems too. Mohanty et al. (2013) evaluated the performances of the finite difference groundwater model MODFLOW and the computational model artificial neural network (ANN) in the simulation of groundwater level in an alluvial aquifer system. They compared the results with field observed data and found that the numerical model is suitable for long-term predictions, whereas the ANN model is appropriate for short-term

applications. Serrano (2013) illustrated the use of Adomian's decomposition method to solve a regional groundwater in an unconfined aquifer bounded by the main stream on one side, two tributaries on two sides, and an impervious boundary on the other side. He demonstrated an application to an aquifer bounded by four streams with a deep excavation inside where the head was kept constant. Jafari et al. (2016) incorporated Terzaghi's theory of one-dimensional consolidation with MODFLOW to evaluate groundwater flow and land subsidence due to heavy pumping in a basin aquifer in Iran. So far, many computer

codes developed based on either FDMs (e.g., FTWORK and MODFLOW), FEMs (e.g., AQUIFEM-N, BEMLAP, FEMWATER, and SUTRA) or boundary element methods (e.g., BEMLAP) had been employed to simulate a variety of groundwater flow problems (Loudyi et al., 2007).

On the other hand, analytical solutions are convenient and powerful tools to explore the physical insight of groundwater flow systems. The head solution is capable of predicting the spatiotemporal distribution of the drawdown at any location within the

simulation time and the *SDR* solution can estimate the stream filtration rate at any instance at a specific location in the groundwater flow system. Thus, the development of analytical models for describing the groundwater flow in a heterogeneous aquifer with irregular outer boundaries and subject to various types of boundary condition is of practical use from an engineering viewpoint. Kuo et al. (1994) applied the image well theory and Theis' equation to estimate transient drawdown in an aquifer with irregularly shaped boundaries. However, the number of the image wells should be largely increased if the



aquifer boundary is asymmetric and rather irregular. Insufficient number of the image wells might result in poor results or even divergence (Matthews et al., 1954). Read and Volker (1993) presented analytical solutions for steady seepage through hillsides with arbitrarily flow boundaries. They used the least squares method to estimate the coefficients in a series expansion of the Laplace equation. Li et al. (1996) extended the results of Read and Volker (1993) in solving the two-dimensional (2D)

groundwater flow in porous media governed by Laplace's equation involving arbitrary boundary conditions. The solution procedure was obtained by means of an infinite series of orthonomal functions. Additionally, they also introduced a method, called image-recharge method, to establish the recurrence relationship of the series coefficients. Patel and Serrano (2011) solved nonlinear boundary value problems of multidimensional equations by Adomian's method of decomposition for groundwater flow in irregularly shaped aquifer domains. Currently, Huang et al. (2016) presented 3D analytical solutions for

hydraulic head distributions and $SDR$s induced by a radial collector well in a rectangular confined or unconfined aquifer bounded by two parallel streams and no-flow boundaries.

Groundwater pumping near a stream in a fluvial aquifer may cause the dispute of stream water right, impact of aquatic ecosystem in stream, as well as water allocation or management problems for agriculture, industry, and municipality. The impacts of groundwater extraction by wells should therefore be thoroughly investigated before pumping. This paper develops

a 2D mathematical model for describing the groundwater flow in an approximately L-shaped fluvial aquifer which is very close to the case of numerical simulations reported in Kihm et al. (2007). The aquifer is divided into two rectangular sub-regions. The aquifer in each sub-region is homogeneous but anisotropic in the horizontal plane with the principal direction aligned with the border of the sub-region. Three types of boundary conditions including constant-head, linearly varying head, and no-flow are adopted to reflect the physical reality at the outer boundaries of the problem domain. A steady-state solution

is first developed to represent the hydraulic head distribution within the aquifer before pumping. The transient head solution of the model is then obtained using the Fourier finite sine and cosine transforms and the Laplace transform. The Stehfest algorithm is then taken to inverse Laplace-domain solution for the time-domain results. The software MODFLOW-2005 for the simulation of the 3D groundwater flow in L-shaped heterogeneous aquifer is used to check the accuracy of the present head solutions. The $SDR$ solution is also derived based on the head solution and Darcy's law and then used to evaluate the

contribution of filtration water from each of two streams toward the pumping well.

## 2 Methodology

Figure 1 shows a fluvial plain located in Yongpoong 2 Agriculture District, Gyeonggi-Do, Korea reported in Kihm et al. (2007). The west side of the plain is a mountainous area with the formation of exposed impermeable bedrock and the east side has the Poonggye stream which passes the district from the southwest corner toward the northeast. A tributary of Poonggye stream, entering the stream with nearly a right angle, is on the north side of the plain. The Poonggye stream and its tributary are

perennial stream and almost fully penetrate the fluvial aquifer system (Kihm et al., 2007). The width of Poonggye stream is about 15m reported in Rhms (2013).



## 2.1 Conceptual Model

The aquifer in the district is formed by fluvial deposit with a total thickness of $6\ m$, and consists of a clay loam aquitard of a thickness of about $2.5\ m$ underlain by a loamy sand layer of a thickness of about $3.5\ m$. In order to develop an analytical model for solving the groundwater flow, the domain of the aquifer in this study is approximated to be L-shaped, as delineated

in Figure 2. Notice that in Figure 1 the solid line denotes the outer boundary of the L-shaped aquifer in this study while the dashed line represents the simulation area in the work of Kihm et al. (2007). The origin of the coordinate in Figure 2 is at the lower left corner of point A, which is at the intersection of boundary AB (i.e., a part of Poonggye stream) and boundary AG. The boundaries of the aquifer domain along EF and FG are impermeable bedrocks and thus regarded as impermeable boundaries. The annual average heads at points A, B, and D are known to be $5.18m$, $4.06m$ and $5.29m$, respectively, above

the bottom of the aquifer (Kihm et al. 2007). The hydraulic heads along AG and DE are fixed at their average water stages as did in Kihm et al. (2007). The boundaries AB and BD are designated to represent the Poonggye stream and its tributary, respectively. Kihm et al. (2007) fixed the hydraulic heads of Poonggye stream and its tributary at annual average water stages in their numerical simulations. Thus, this study considers that the stream has a perfect hydraulic connection with the aquifer and the stream stage varies linearly with distance. The average stream flow rate of the Poonggye stream with its tributary is

about 100 m³/s reported in Rhms (2013, p. 90). Todd and Mays (2005, p. 232) mentioned that the pumping rate in a shallow well with suction lift less than 7 m may range up to 500 m³/day (0.01 m³/s). Hence, the effect of pumping in a shallow well on the water table of nearby stream is generally negligible. The average depth to the water table from the ground surface is $1.26m$ with a spatial variation between $0.57m$ and $1.95m$ in accordance with the average water stages in the streams AB and BD. This aquifer is divided into two regions, named regions 1 and 2, and the hydraulic heads in these two regions are

respectively expressed as $\phi_1(x, y, t)$ and $\phi_2(x, y, t)$.

## 2.2 Mathematical model

Consider that there are totally $M$ pumping wells in region 1 and $N$ pumping wells in region 2. The coordinates of $k$th well in region 1 is denoted as $(x_{1k}, y_{1k})$ and the associated pumping rate per unit thickness is $Q_{1k}$ $[L^2/T]$. The governing equation describing the 2D hydraulic head distribution in region 1 is expressed as

$$K_{x1}\frac{\partial^2 \phi_1}{\partial x^2} + K_{y1}\frac{\partial^2 \phi_1}{\partial y^2} = S_{s1}\frac{\partial \phi_1}{\partial t} - \sum_{k=1}^{M} Q_{1k}\delta(x - x_{1k})\delta(y - y_{1k})$$

$$0 \leq x \leq l_1, 0 \leq y \leq b_1 \tag{1}$$

Similarly, the 2D hydraulic head distribution in region 2 for the $l$ th well located at $(x_{2l}, y_{2l})$ with a pumping rate per unit thickness of $Q_{2l}$ $[L^2/T]$ is

$$K_{x2}\frac{\partial^2 \phi_2}{\partial x^2} + K_{y2}\frac{\partial^2 \phi_2}{\partial y^2} = S_{s2}\frac{\partial \phi_2}{\partial t} - \sum_{l=1}^{N} Q_{2l}\delta(x - x_{2l})\delta(y - y_{2l})$$



$$l_2 \leq x \leq l_1, b_1 \leq y \leq b_2 \tag{2}$$

where $S_s\,[L^{-1}]$ represents the specific storage, $K_x\,[L/T]$ and $K_y\,[L/T]$ are the hydraulic conductivities in $x$- and $y$-direction, respectively. The symbol $\delta$ represents one dimensional (1D) Dirac's delta function.

The boundary conditions for region 1 are expressed as:

$$\phi_1(0, y) = h_1 \text{ for AG} \tag{3}$$

$$\phi_1(l_1, y) = h_3 + \frac{h_2 - h_3}{b_2} y \text{ for BC} \tag{4}$$

$$\phi_1(x, 0) = h_1 + \frac{h_3 - h_1}{l_1} x \text{ for AB} \tag{5}$$

$$\frac{\partial \phi_1}{\partial y}(x, b_1) = 0 \text{ for GF} \tag{6}$$

Similarly, the boundary conditions for flow in region 2 are

$$\frac{\partial \phi_2}{\partial x}(l_2, y) = 0 \text{ for EF} \tag{7}$$

$$\phi_2(l_1, y) = h_3 + \frac{h_2 - h_3}{b_2} y \text{ for CD} \tag{8}$$

$$\phi_2(x, b_2) = h_2 \text{ for DE} \tag{9}$$

The continuity requirements of hydraulic head and flux along the interface CF are respectively

$$\phi_1(x, b_1) = \phi_2(x, b_1) \tag{10}$$

and

$$K_{y1} \left.\frac{\partial \phi_1}{\partial y}\right|_{y=b_1} = K_{y2} \left.\frac{\partial \phi_2}{\partial y}\right|_{y=b_1} \tag{11}$$

In order to express the solution in dimensionless form, the following dimensionless variables or parameters are introduced: $\phi_1^* = (\phi_1 - h_1)/h_1$, $\phi_2^* = (\phi_2 - h_1)/h_2$, $t^* = K_{y1}t/S_{s1}b_2^2$, $\kappa_1 = K_{x1}/K_{y1}$, $\kappa_2 = K_{x2}/K_{y2}$, $x^* = x/l_1$, $y^* = y/b_2$, $b_1^* = b_1/b_2$, $l_2^* = l_2/l_1$, $Q_{1k}^* = b_2^2 Q_{1k}/K_{y1}h_1$ and $Q_{2l}^* = b_2^2 Q_{2l}/K_{y2}h_2$ where $\phi_1^*$ and $\phi_2^*$ stand for the dimensionless hydraulic heads in regions 1 and 2, respectively; $t^*$ refers to the dimensionless time during the test; $\kappa_1$ and $\kappa_2$ represent the anisotropic ratio of hydraulic conductivity in region 1 and 2, respectively; $x^*$ and $y^*$ denote the dimensionless coordinates.

## 2.3 Steady-state solution for hydraulic head distribution

In order to compare the steady-state simulations of Kihm et al. (2007) without pumping, the steady-state solution for the hydraulic head distribution in the L-shaped aquifer is developed. Detailed derivation for the analytical solutions in steady state for regions 1 and 2 is given in Appendix A, and the results are expressed respectively as (Chu et al., 2012)



$$\phi_1^*(x^*, y^*) = \sum_{m=1}^{\infty} \Delta_1 [C_{1m} E_1(m, y^*) + F_1(m, y^*)] \sin(\lambda_m x^*) \tag{12}$$

and

$$\phi_2^*(x^*, y^*) = \sum_{n=1}^{\infty} \Delta_2 [D_{2n} E_2(n, y^*) + F_2(n, y^*)] \cos[\alpha_n(x^* - l_2^*)] \tag{13}$$

with

$$E_1(m, y^*) = \frac{e^{\Omega_{1m} y^*} - e^{-\Omega_{1m} y^*}}{e^{\Omega_{1m} b_1^*} - e^{-\Omega_{1m} b_1^*}} \tag{14}$$

$$F_1(m, y^*) = \frac{1}{\lambda_m} (-1)^{m+1} (h_{31}^* + h_{23}^* y^*) \tag{15}$$

$$E_2(n, y^*) = e^{-\Omega_{2n} y^*} - e^{\Omega_{2n}(y^* - 2)} \tag{16}$$

$$F_2(n, y^*) = \frac{(-1)^{n-1}}{\alpha_n} (H_{31}^* + H_{23}^* y^*) \tag{17}$$

where the symbols and dimensionless variables $\Delta_1$, $\Delta_2$, $\lambda_m$, $\alpha_n$, $\Omega_{1m}$, $\Omega_{2n}$, $h_{31}^*$, $h_{23}^*$, $H_{23}^*$ and $H_{31}^*$ are defined in Table 1. The coefficients $C_{1m}$ and $D_{2n}$ can be determined simultaneously by the continuity conditions of hydraulic head and flux along the interface CF. The results are denoted as follows:

$$C_{1m} = \frac{\Delta_2}{2} \frac{K_{y2}}{K_{y1}} \frac{h_2}{h_1} \sum_{n=1}^{\infty} \left[ D_{2n} \frac{E_2'(n, y^*)}{E_1'(m, y^*)} \Big|_{y^* = b_1^*} + \frac{F_2'(n, y^*)}{E_1'(m, y^*)} \Big|_{y^* = b_1^*} \right] G_1(m, n) - \frac{F_1'(m, y^*)}{E_1'(m, y^*)} \Big|_{y^* = b_1^*} \tag{18}$$

and

$$D_{2n} = \frac{1}{\Delta_2} \frac{h_1}{h_2} \sum_{m=0}^{\infty} \left[ C_{1m} \frac{E_1(m, b_1^*)}{E_2(n, b_1^*)} + \frac{F_1(m, b_1^*)}{E_2(n, b_1^*)} \right] G_2(m, n) - \frac{F_2(n, b_1^*)}{E_2(n, b_1^*)} \tag{19}$$

with

$$E_1'(m, y^*) = \frac{\partial E_1(m, y^*)}{\partial y^*} \tag{20}$$

$$E_2'(n, y^*) = \frac{\partial E_2(n, y^*)}{\partial y^*} \tag{21}$$

$$F_1'(m, y^*) = \frac{\partial F_1(m, y^*)}{\partial y^*} \tag{22}$$

$$F_2'(n, y^*) = \frac{\partial F_2(n, y^*)}{\partial y^*} \tag{23}$$

$$G_1(m, n) \frac{\int_{l_2^*}^{1} \sin(\lambda_m x^*) \cos[\alpha_n(x^* - l_2^*)] dx}{\int_0^1 \sin^2(\lambda_m x^*) dx} \tag{24}$$

$$G_2(m, n) \frac{\int_{l_2^*}^{1} \sin(\lambda_m x^*) \cos[\alpha_n(x^* - l_2^*)] dx}{\int_{l_2^*}^{1} \cos^2[\alpha_n(x^* - l_2^*)] dx} \tag{25}$$





### 2.4 Transient solution for hydraulic head distribution

The solution of the model for transient hydraulic head distribution with the previous steady-state solution as the initial condition is developed via the methods of finite sine transform, finite cosine transform and Laplace transform. The detailed derivation for transient solution is given in Appendix B and the results of the dimensionless hydraulic heads in Laplace domain for regions 5 1 and 2 are respectively

$$\tilde{\phi}_1^*(x^*, y^*, p) = \delta_1 \sum_{i=1}^{\infty} [w_{1i}^* TE_1(i, y^*, p) + T_1(i, y^*, p) + T_2(i, y^*, p) + SQ_1(i, y^*, p)] \sin(\lambda_i x^*) \tag{26}$$

and

$$\tilde{\phi}_2^*(x^*, y^*, p) = \delta_2 \sum_{j=1}^{\infty} [w_{2j}^* TE_2(j, y^*, p) + T_4(j, y^*, p) + T_5(j, y^*, p) + SQ_2(j, y^*, p)] \cos[\alpha_j(x^* - l_2^*)] \tag{27}$$

with

$$TE_1(i, y^*, p) = \frac{e^{\mu_i(y^* - b_1^*)} - e^{-\mu_i(y^* + b_1^*)}}{1 - e^{-2\mu_i b_1^*}} \tag{28}$$

$$T_1(i, y^*, p) = \frac{1}{\mu_i p} [\theta_1^2 \lambda_i (-1)^i][h_{31}^* e^{-\mu_i y^*} + (h_{23}^* y^* - h_{31}^*)] - h_{31}^* \frac{(-1)^i}{\lambda_i} e^{-\mu_i y^*} \tag{29}$$

$$T_2(i, y^*, p) = -C_{1m} \left[\frac{1}{2} - \frac{\sin(2\lambda_i)}{4\lambda_i}\right] \left[\frac{-e^{-\Omega_{1i}(y^* - b_1^*)} + e^{\Omega_{1i}(y^* - b_1^*)}}{(1 - e^{2\Omega_{1i} b_1^*})(\Omega_{1i}^2 - \mu_i^2)} + \frac{\delta_1(-1)^i}{\mu_i^2}(h_{23}^* y^* + h_{31}^* - h_{31}^* e^{-\mu_i y^*})\right] \tag{30}$$

$$SQ_1(i, y^*, p) = \begin{cases} \frac{1}{2\mu_i p} \sum_{k=1}^{M} Q_{1k}^* \sin(\lambda_i x_{1k}^*) \frac{1}{1 - e^{-\mu_i b_1^*}} \begin{bmatrix} e^{\mu_i(y_{1k}^* - 2b_1^*)} + e^{\mu_i(y^* - y_{1k}^* - b_1^*)} \\ + e^{\mu_i(y_{1k}^* - y^*)} - e^{\mu_i(y^* - y_{1k}^* - 2b_1^*)} \\ -e^{\mu_i(y_{1k}^* - y^* - b_1^*)} - e^{-\mu_i y_{1k}^*} \end{bmatrix}, y^* > y_{1k}^* \\ \frac{1}{2\mu_i p} \sum_{k=1}^{M} Q_{1k}^* \sin(\lambda_i x_{1k}^*) \frac{1}{1 - e^{-\mu_i b_1^*}} \begin{bmatrix} e^{\mu_i(y^* - y_{1k}^*)} + e^{\mu_i(y_{1k}^* - 2b_1^*)} \\ -e^{\mu_i(y^* + y_{1k}^* - 2b_1^*)} - e^{-\mu_i y_{1k}^*} \end{bmatrix}, y^* < y_{1k}^* \end{cases} \tag{31}$$

$$TE_2(j, y^*, p) = \frac{e^{-\theta_j y^*} - e^{\theta_j(y^* - 2)}}{e^{-\theta_j b_1^*} - e^{\theta_j(b_1^* - 2)}} \tag{32}$$

$$T_4(j, y^*, p) = \left[\frac{\theta_2^2 \alpha_j (-1)^j}{p} + \frac{S_{s2} K_{y1} (-1)^j}{S_{s1} K_{y2} \alpha_j}\right] \left[\frac{H_{31}^* + H_{23}^* y^* - H_{21}^* e^{\theta_j(y^* - 1)}}{\theta_j^2}\right] + \frac{H_{21}^*}{\alpha_j} \frac{(-1)^j}{p} e^{\theta_j(y^* - 1)} \tag{33}$$

$$T_5(j, y^*, p) = -D_{2n} \frac{S_{s2} K_{y1}(1 - l_2^*)}{2 S_{s1} K_{y2}} \left(\frac{e^{\Omega_{2j}(y^* - 2)} - e^{-\Omega_{2j} y^*}}{\theta_j^2 - \Omega_{2j}}\right) \tag{34}$$

$$SQ_2(j, y^*, p) = \begin{cases} \frac{1}{2\theta_j p} \sum_{l=1}^{N} Q_{2l}^* \cos(\alpha_j x_{2l}^*) \left(e^{-\theta_j(y^* - y_{2l}^*)} - e^{-\theta_j(2 - y^* - y_{2l}^*)}\right), y^* > y_{2j}^* \\ \frac{1}{2\theta_j p} \sum_{l=1}^{N} Q_{2l}^* \cos(\alpha_j x_{2l}^*) \left(e^{\theta_j(y^* - y_{2l}^*)} - e^{-\theta_j(2 - y^* - y_{2l}^*)}\right), y^* < y_{2j}^* \end{cases} \tag{35}$$

where $p$ is the Laplace variable and the symbols or dimensionless parameters $\delta_1$, $\delta_2$, $\alpha_j$, $\lambda_i$, $\mu_i$, $\theta_1$, $\theta_2$, $\theta_j$, $\Omega_{1i}$, $\Omega_{2j}$ and $H_{21}^*$ are introduced in Table 1.



The coefficients in Eqs. (26) and (27) are obtained via continuity requirements for the hydraulic head and flow flux at the interface CF. They can be solved simultaneously based on the following two equations

$$w_{1i}^* = \frac{K_{y2}}{K_{y1}}\frac{h_2}{h_1}G_1(i,j)\sum_{i=1}^{\infty}\left.\frac{w_{2j}^*TE_2'(j,y^*,p)+T_4'(j,y^*,p)+T_5'(j,y^*,p)+SQ_2'(j,y^*,p)}{TE_1'(j,y^*,p)}\right|_{y^*=b_1^*} + \sum_{i=1}^{\infty}\left.\frac{-T_1'(j,y^*,p)+T_2'(j,y^*,p)+SQ_1'(j,y^*,p)}{TE_1'(j,y^*,p)}\right|_{y^*=b_1^*} \quad (36)$$

and

$$w_{2j}^* = \frac{h_1}{h_2}G_2(j,i)\sum_{j=1}^{\infty}\left.\frac{w_{1i}^*TE_1(i,y^*,p)+T_1(i,y^*,p)-T_2(i,y^*,p)+SQ_1(i,y^*,p)}{TE_2(j,y^*,p)}\right|_{y^*=b_1^*} - \sum_{j=1}^{\infty}\left.\frac{T_4(j,y^*,p)+T_5(j,y^*,p)+SQ_2(j,y^*,p)}{TE_2(j,y^*,p)}\right|_{y^*=b_1^*} \quad (37)$$

with

$$TE_1'(i,y^*,p) = \frac{\partial TE_1(i,y^*,p)}{\partial y^*} \quad (38)$$

$$TE_2'(j,y^*,p) = \frac{\partial TE_2(j,y^*,p)}{\partial y^*} \quad (39)$$

$$T_1'(i,y^*,p) = \frac{\partial T_1(i,y^*,p)}{\partial y^*} \quad (40)$$

$$T_2'(i,y^*,p) = \frac{\partial T_2(i,y^*,p)}{\partial y^*} \quad (41)$$

$$T_4'(j,y^*,p) = \frac{\partial T_4(j,y^*,p)}{\partial y^*} \quad (42)$$

$$T_5'(j,y^*,p) = \frac{\partial T_5(j,y^*,p)}{\partial y^*} \quad (43)$$

$$SQ_1'(i,y^*,p) = \frac{\partial SQ_1(i,y^*,p)}{\partial y^*} \quad (44)$$

$$SQ_2'(j,y^*,p) = \frac{\partial SQ_2(j,y^*,p)}{\partial y^*} \quad (45)$$

The coefficient $w_{2j}^*$ can be determined by substituting Eq. (36) into Eq. (37), the $w_{1i}^*$ can then be obtained once $w_{2j}^*$ is known. The hydraulic head distributions in real time domain can be obtained by applying a numerical Laplace inversion scheme, called the Stehfest algorithm (Stehfest, 1970), to Eqs. (26) and (27).

## 2.5 Stream depletion rate

Pumping in an aquifer near a stream will produce filtration water from the stream toward the well (Yeh et al., 2008). Water extracted from the pumping well comes from different sources such as nearby streams and aquifer storage. The extraction rate from the stream is referred to as stream depletion rate (*SDR*) and that from the aquifer storage is storage release rate (*SRR*). The dimensionless solutions of *SDR* in Laplace domain from the stream reaches AB and BD, denoted respectively as $\widetilde{SDR}_A$





and $\widehat{SDR}_B$, can be estimated by taking the derivatives of Eqs. (26) and (27) with respect to $y$ and $x$, respectively, then integrating along the reaches as:

$$\widehat{SDR}_A = \frac{q_A}{Q} = -\frac{1}{Q}\left(\int_0^{l_1} K_h \left.\frac{\partial \bar{\phi}_1(x,y,p)}{\partial y}\right|_{y=0} dx + \int_0^{l_1} K_h \left.\frac{\partial \bar{\phi}_2(x,y,p)}{\partial y}\right|_{y=0} dx\right) \tag{46}$$

and

$$\widehat{SDR}_B = \frac{q_B}{Q} = \frac{1}{Q}\left(\int_0^{b_2} K_h \left.\frac{\partial \bar{\phi}_1(x,y,p)}{\partial x}\right|_{x=l_1} dy + \int_0^{b_2} K_h \left.\frac{\partial \bar{\phi}_2(x,y,p)}{\partial x}\right|_{x=l_1} dy\right) \tag{47}$$

where $K_h$ is the equivalent horizontal hydraulic conductivity, $q_A$ and $q_B$ are the filtration rates from streams AB and BD, respectively. Assuming only horizontal flow in the aquifer, the equivalent horizontal hydraulic conductivity $K_h$ for layered aquifer system is estimated based on the following formula (Schwartz and Zhang, 2003):

$$K_h = \sum_i^m B_i K_i / \sum_i^m B_i \tag{48}$$

where $K_i$ is the hydraulic conductivity in the horizontal direction for layer $i$ and $B_i$ is the thickness of layer $i$. The dimensionless total $SDR$ in Laplace domain from the streams can therefore be written as:

$$\widehat{SDR}_T = \widehat{SDR}_A + \widehat{SDR}_B \tag{49}$$

The dimensionless time domain solutions for $SDR_A$, $SDR_B$ and $SDR_T$ can also be evaluated by the Stehfest algorithm. The dimensionless $SRR$ representing the storage release rate due to the compression of aquifer matrix and the expansion of groundwater in the pore space can be written as:

$$SRR = Q^* - SDR_T \tag{50}$$

## 3 Solution validation and applications

### 3.1 Solution validation by MODFLOW-2005

A 3D numerical model for investigating the hydraulic head distribution within the L-shaped fluvial aquifer of Yongpoong 2 Agriculture District is developed using MODFLOW-2005. MODFLOW is a widely used finite-difference model developed by U.S. Geological Survey for the simulation of 3D groundwater flow problems under various hydrogeological conditions (USGS, 2005). As shown in Figure 1, region 1 has an area of $852m \times 222m$ (i.e., $l_1 \times b_1$) while the area of region 2 is $297m \times 183m$ (i.e., $(l_{1-}l_2) \times (b_2 - b_1)$). Thus, the total area of these two regions is $243495\ m^2$ which is close to the area of the fluvial aquifer ($246500m^2$) reported in Kihm et al. (2007). In the simulation of MODFLOW-2005, the plane of the L-shaped aquifer is discretized with a uniform cell size of $3m \times 3m$. The aquifer thickness is $6m$ and divided into two layers. The upper loam layer is $2.5m$ and lower sand layer $3.5m$. Within the aquifer domain, there is totally 54110 cells while the numbers of cell are 42032 and 12078 respectively for region 1 and region 2. The types of outer boundary specified for the L-





shaped aquifer are the same as those defined in the mathematical model. The hydraulic heads along AG and DE are respectively $h_1 = 5.18m$ and $h_2 = 5.29m$ and the head at point B is $h_3 = 4.06m$. The hydraulic conductivities for the upper and lower layers are $3 \times 10^{-6} m/s$ and $2 \times 10^{-4} m/s$, respectively, considered in the simulations of MODFLOW-2005. On the other hand, the equivalent hydraulic conductivity $K_h$ calculated by Eq. (48) as $1.2 \times 10^{-4} m/s$ is used by the present solution. The

specific storage of the aquifer in both regions 1 and 2 is $10^{-4} m^{-1}$. Consider that the pumping well $P_w$ is located at $(609m, 9m)$ in region 1 shown in Figure 2 with a rate of $120 m^3/day$ for one year pumping. Figure 3 shows the hydraulic head distributions denoted as the solid line predicted by the present solution of Eqs. (26) and (27) and represented by the dotted line given by the MODFLOW-2005. The figure indicates a good agreement between these two predicted results, indicating that the present solution gives a fairly good prediction. The largest relative deviation of 2.1% occurs near the no-flow boundary FG where the

predicted head is about $4.6m$.

### 3.2 Steady-state head distribution without pumping in Yongpoong 2 Agriculture District and impact of aquifer anisotropy

Kihm et al. (2007) reported the steady-state hydraulic head distribution, shown in Figure 4 by the dashed line, for the FEM simulation without groundwater pumping in the two-layered L-shaped aquifer. Figure 4 also shows the steady-state head

distribution, denoted as the solid line, predicted by the present solution of Eqs. (11) and (12) for the L-shaped aquifer with $K_{x1} = K_{y1} = K_{x2} = K_{y2} = 1.2 \times 10^{-4} m/s$ (i.e., $\kappa_1 = \kappa_2 = 1$) evaluated based on Eq. (48) and other aquifer properties mentioned in section 3.1. The contour lines of the head distribution are nearly parallel to the boundary AG and perpendicular to the boundary FG in the region $x \le 200m$. Moreover, the predicted heads within the regions between $500m \le x \le 852m$ and $0m \le y \le 200m$ are reasonably close to the FEM results, which range from $4.3m$ to $4.7m$ as shown in Figure 4. The

groundwater flows toward point B since it has the lowest water table within the problem domain.

Figure 5 shows the contour lines of the hydraulic head distribution for isotropic case of $\kappa_1 = \kappa_2 = 1$ by the solid line and for anisotropic cases of $\kappa_1 = \kappa_2 = 4$ represented by the dashed dot line and $\kappa_1 = \kappa_2 = 0.25$ by the dashed line. In these three cases, the head distributions are significantly different in the region where $x \le 600m$ for the head ranging from $5m$ to $4.6m$. The largest head difference occurs near the upper boundary FG, reflecting the effects of no-flow condition and aquifer

anisotropy on the flow pattern within this area.

### 3.3 Spatial head distributions due to pumping simulated by Kihm et al. (2007) and present solution after one year pumping

Note that Figure 3 shows the spatial head distributions in the L-shaped aquifer predicted by the present solution and the MODFLOW-2005 for one-year pumping at well $P_w$ located at $(609m, 9m)$ with a rate of $120 m^3/day$. In fact, Kihm et al.

(2007) reported their FEM simulations for head distributions, groundwater flow velocity, and land displacement for one-year pumping at the well $P_w$ with the same pumping rate mentioned above. They referred the simulated head results as initial steady-state distributions for the case of no pumping and final steady-state distributions for the case after one-year pumping.



The aquifer configuration in their FEM simulations and the simulated head distributions denoted as dashed line are also demonstrated in Figure 3. The figure indicates that the present solution gives good predicted head contours near the pumping well and reasonably good result for the head distribution in region 1 as compared to those given by Kihm et al. (2007). Notice that the pumping well is very close to the stream boundary AB, which is the main stream in that area and provides a large

amount of filtration water to the well. Hence, it seems that the groundwater flows in the region 1 for $x \leq 300m$ and in the region 2 for $y \geq 200m$ are both far away from the well and almost not influenced by the pumping.

Three piezometers $O_1$, $O_2$ and $O_3$ were respectively installed at $(597m, 25m)$, $(594m, 48m)$ and $(597m, 204m)$ mentioned in Kihm et al. (2007) and indicated in Figure 2. Note that $O_1$ was installed near the stream AB while $O_3$ was far away from the stream but close to the impermeable upper boundary. Figure 6 shows the temporal distributions of hydraulic head measured at

these three piezometers (i.e., $H_{iM}$, $i = 1, 2, 3$) and predicted by the FEM simulations (Kihm et al., 2007) (i.e., $H_{iF}$) and present solution (i.e., $H_{iA}$). This figure indicates that the hydraulic heads predicted by the present solution has a good agreement with those simulated by Kihm et al. (2007). The relative differences of predicted hydraulic head between FEM simulations and present solution are all less than 0.8% at these three piezometers over the entire pumping period. In addition, the largest relative differences between measured heads and predicted heads by the present solution at $O_1$ to $O_3$ are respectively 1.64%, 1.74%

and 0.62%. The hydraulic head at $O_1$ declines greater than those at $O_2$ and $O_3$ whereas the former is located closer to the pumping well $P_w$. Because $P_w$ is very near the stream, the extracted water will be quickly contributed from the stream and therefore the drawdown at $O_1$ will be soon stabilized. Figure 6 indicates that the hydraulic heads at $O_1$ - $O_3$ predicted by the present solution reach steady state after $t = 100$ days, 220 days and 290 days, respectively.

### 3.4 Stream filtration in fluvial aquifer systems

Stream filtration can be considered as a problem associated with the interaction between the groundwater and surface water. The pumped water originated from the nearby stream is commonly supplied for irrigation, municipalities, and rural homes. In stream basins with several tributaries, pumping wells are often installed adjacent to the confluence of two tributaries in fluvial aquifers (Lambs, 2004).

It is of practical interest to know the temporal *SDR* distributions from both streams in the Yongpoong area when subject to

pumping at $P_w$ under a rate of 120 m³/day. The distances from $P_w$ to the streams AB and BD are respectively $9m$ and $243m$. Figure 7 shows the temporal *SDR* distribution from each stream, indicating that $SDR_A$ (i.e., *SDR* from stream AB) is significantly larger than $SDR_B$ (*SDR* from stream BD) all the time. The steady-state values for $SDR_A$ and $SDR_B$ are respectively 0.81 and 0.19 when $t \geq 220$ day. This is due to the fact that pumping well is closer to stream AB than stream BD and therefore water contributing to the pumping well from stream AB is much more than from stream BD. Figure 7 also shows that the $SDR_T$

($SDR_A+SDR_B$) is zero and the aquifer storage release rate *SRR* is unity when $t \leq 0.01$ day, indicating that the well discharges totally from the aquifer storage at early time. Once the drawdown cone reaches the stream, the $SDR_T$ increases quickly with time while the *SRR* decreases continuously over the entire pumping period. It is interesting to mention that $SDR_T$ starts to



contribute more water to the pumping than $SRR$ when $t \geq 5$ day. Finally, the $SDR_T$ reaches unity and the $SRR$ equals zero after $t \geq 220$ days, indicating that the aquifer system approaches steady state and all the extraction water comes from the streams.

**3.5 Determination of well location for a specific SDR in a L-Shaped aquifer**

It is of interest to mention that the present solution can be a preliminary design tool in determining the location of a pumping well in an L-shaped aquifer if the amounts of $SDR_A$ and $SDR_B$ had been determined by water authority or based on water right. Driscoll (1986, p. 615) mentions that a well shall be installed at least $45.7m$ from areas of spray materials, fertilizers or chemicals that contaminate the soil or groundwater. Hence, the distance from the pumping well to the stream is considered at least 50m.

Two specific values of $SDR_A$, 0.65 and 0.75, are considered for a water supply rate of $120 m^3/day$. The present solution is employed to determine the well locations in order to meet the water need for irrigation in an L-shaped aquifer. Since stream AB is the main stream, it is better to extract more filtration water from it than from its tributary, stream BD. Therefore, four trial pumping wells, $P_1$ to $P_4$, are considered to install for the distances $d_{B1}$ to $d_{B4}$ of $50m$, $100m$, $150m$ and $200m$, respectively, with $d_A = 50m$ indicated in Figure 2. The steady-state $SDR_A$ predicted by the present solution for pumping at each of $P_1$ to $P_4$ with an extraction rate of $120 m^3/day$ is respectively 0.52, 0.61, 0.74, and 0.78. The least-squares equation with second degree polynomial for $d_B$ in terms of $SDR_A$ is $d_B = 591.4x^2 - 239.9x + 18.01$ with $x$ representing the $SDR_A$. The estimated $d_B$ for $SDR_A = 0.65$ and 0.75 by the regression equation are 111.95m and 170.76m, respectively. The predicted $SDR_A$ from the present solution is 0.63 and its relative difference is 3.0% for pumping well located at (111.95m, 50m). On the other hand, the predicted $SDR_A$ from the present solution is 0.76 and its relative difference is 1.3% for well at (170.76m, 50m). In these two cases, we demonstrate that the present solution can be used as a design tool to determine the well location for a specific amount of filtration water from nearby streams in an L-shaped aquifer.

**4 Conclusions**

A new analytical model has been developed to analyze the 2D hydraulic head distributions with/without pumping in a heterogeneous and anisotropic aquifer for an L-shaped domain bounded by two streams with linearly varying hydraulic heads. Method of domain decomposition is used to divide the aquifer into two regions for the development of semi-analytical solution. Steady-state solution is first derived and used as the initial condition for the L-shaped aquifer system before pumping. The Laplace-domain solution of the model for transient head distribution in the aquifer subject to pumping is developed using the Fourier finite sine and cosine transform and the Laplace transform. The solution for $SDR$ describing filtration rate from two streams in an L-shaped aquifer is developed based on the head solution and Darcy's law. The Stehfest algorithm is then adopted to evaluate the time-domain results for both head and $SDR$ solutions in Laplace domain.



The 3D finite difference model MODFLOW-2005 is first used to check the accuracy of hydraulic head predictions by the present solution for the L-shaped two-layered aquifer system. The hydraulic head distributions predicted by present solutions agree fairly well over the entire aquifer except the heads nearing the no-flow boundary. The solution for hydraulic head distribution in the L-shaped aquifer without pumping has been used to investigate the effect of anisotropic ratio ($K_x/K_y$) on

the steady-state flow system. It is interesting to note that the flow pattern in terms of lines of equal hydraulic head is strongly influenced by the value of anisotropic ratio for the region near the turning point of the L-shaped aquifer. The transient solution for head distribution is employed to simulate the head distribution induced by pumping in the aquifer within the agriculture area of Gyeonggi-Do, Korea. The aquifer is approximated as L-shaped in this study. The simulation results indicate that the largest relative difference in predicted temporal head distributions at three piezometers by the present solution and Kihm et

al.'s (2007) FEM simulation is less than 1.74%, implying that the effects of unsaturated flow and land deformation on the groundwater flow in Yongpoong aquifer are small and may be negligible.

The *SDR* solution is first used to evaluate the steady-state *SDR* from each of the nearby streams for Yongpoong aquifer subject to a specific pumping rate. The solution is also employed to determine the temporal contribution rates from the aquifer storage and the streams toward the extraction well. Then the solution is used as a design tool to determine the well location with a

specific pumping rate for required amounts of *SDR* from nearby streams. Two cases are provided with four trial pumping wells assumed at distances at least 50m from the streams. A quadratic equation is considered with the dependent variable representing the distance ($d_B$) from the trial well to one of the stream (stream BD) and the independent variable denoted as the estimated *SDR* (from stream AB, the main stream in Yongpoong area) predicted by the present solution at the trial pumping wells. The quadratic equation with coefficients estimated by the least squares approach is then used to determine the pumping well

location for a required *SDR* from nearby streams in an L-shaped aquifer. In the case studies, the estimated *SDR* by the present solution at the well location predicted by the regression equation yields about 3.0% relative error for the required *SDR* of 0.63 and 1.3% relative error for that of 0.76. These results indicate that the present solution can be used as a preliminary design tool in determining the well location for a required amount of *SDR*.

**Acknowledgements**

This study was partly supported by the grants from Taiwan Ministry of Science and Technology under the contract number MOST 105-2221-E-009-043-MY2.

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

## Appendix A. Steady-state solution for flow in an L-shaped aquifer without pumping

On the basis of dimensionless variables and parameters defined in section 2.2, Eqs. (1) and (2) can be written respectively as

$$\kappa_1 \frac{b_2^2}{l_1^2} \frac{\partial^2 \phi_1^*}{\partial x^{*2}} + \frac{\partial^2 \phi_1^*}{\partial y^{*2}} = \frac{\partial \phi_1^*}{\partial t^*} - \sum_{k=1}^{M} Q_{1k}^* \delta(x^* - x_{1k}^*)\delta(y^* - y_{1k}^*)$$

$0 \leq x^* \leq l_1^*, 0 \leq y^* \leq b_1^*$ (A1)

and

$$\kappa_2 \frac{b_2^2}{l_1^2} \frac{\partial^2 \phi_2^*}{\partial x^{*2}} + \frac{\partial^2 \phi_2^*}{\partial y^{*2}} = \frac{S_{s2}}{S_{s1}} \frac{K_{y1}}{K_{y2}} \frac{\partial \phi_2^*}{\partial t^*} - \sum_{l=1}^{N} Q_{2l}^* \delta(x^* - x_{2l}^*)\delta(y^* - y_{2l}^*)$$

$0 \leq x^* \leq l_2^*, b_1^* \leq y^* \leq 1$ (A2)

The dimensionless boundary conditions for region 1 can be expressed as:

$\phi_1^*(0, y^*) = 0$ for boundary AG (A3)

$\phi_1^*(1, y^*) = h_{31}^* + h_{23}^* y^*$ for boundary BC (A4)

$\phi_1^*(x^*, 0) = h_{31}^* x^*$ for boundary AB (A5)

$\frac{\partial \phi_1^*}{\partial y^*}(x^*, b_1^*) = 0$ for GF (A6)

and for region 2 are

$\frac{\partial \phi_2^*}{\partial x^*}(l_2^*, y^*) = 0$ for boundary EF (A7)





$$\phi_2^*(1, y^*) = h_{31}^* + h_{23}^* y^* \text{ for boundary CD} \tag{A8}$$

$$\phi_2^*(x^*, 1) = h_{21}^* \text{ for boundary DE} \tag{A9}$$

The continuity requirements of hydraulic head and flux at the region interface in dimensionless form are respectively expressed as

$$h_1 \phi_1^*(x^*, b_1^*) = h_2 \phi_2^*(x^*, b_1^*) \text{ for segment CF} \tag{A10}$$

and

$$K_{y1} h_1 \left. \frac{\partial \phi_1^*}{\partial y^*} \right|_{y^* = b_1^*} = K_{y2} h_2 \left. \frac{\partial \phi_2^*}{\partial y^*} \right|_{y^* = b_1^*} \text{ for segment CF} \tag{A11}$$

The steady-state solution for groundwater flow in an L-shaped aquifer without pumping can be solved after removing the source/sink term in Eqs. (A1) and (A2). Multiplying Eq. (A1) by $sin(\lambda_m x^*)$ and integrating it for $x^*$ from 0 to 1 in region 1 with boundary conditions Eqs. (A3) and (A4), Eq. (A1) is then transformed to

$$\Omega_{1m}^2 \bar{\phi}_1^* - \frac{\partial^2 \bar{\phi}_1^*}{\partial y^{*2}} = -\kappa_1 \frac{b_2^2}{l_1^2} \lambda_m (-1)^m (h_{31}^* + h_{23}^* y^*) \tag{A12}$$

with

$$\bar{\phi}_1^* = \int_0^1 \phi_1^* sin(\lambda_m x^*) dx^* \tag{A13}$$

where $\Omega_{1m} = \lambda_m \sqrt{\kappa_1} \, b_2 / l_1$, $\lambda_m = m\pi$ and $m = 1, 2, 3, ....$

Similarly, Eq. (A2) can be transformed via multiplying Eq. (A2) by $cos[\alpha_n(x^* - l_2^*)]$ and integrating it for $x^*$ from $l_2^*$ to 1 in region 2 with boundary conditions Eqs. (A7) and (A8). The result is

$$\Omega_{2n}^2 \bar{\phi}_2^* - \frac{\partial^2 \bar{\phi}_2^*}{\partial y^{*2}} = \kappa_2 \frac{b_2^2}{l_1^2} \alpha_n (-1)^{n-1} (H_{31}^* + H_{23}^* y^*) \tag{A14}$$

with

$$\bar{\phi}_2^* = \int_{l_2^*}^1 \phi_2^* cos[\alpha_n(x^* - l_2^*)] dx^* \tag{A15}$$

where $\Omega_{2n} = \alpha_n \sqrt{\kappa_2} \, b_2 / l_1$, $\alpha_n = (n - 1/2)\pi / (1 - l_2^*)$ and $n = 1, 2, 3, ....$

The general solutions of Eqs. (A12) and (A14) can be written respectively as

$$\bar{\phi}_1^*(m, y^*) = C_{1m} e^{\Omega_{1m} y^*} + C_{2m} e^{-\Omega_{1m} y^*} - \frac{(-1)^m}{\lambda_m} (h_{31}^* + h_{23}^* y^*) \tag{A16}$$

and

$$\bar{\phi}_2^*(n, y^*) = D_{1n} e^{\Omega_{2n} y^*} + D_{2n} e^{-\Omega_{2n} y^*} - \frac{(-1)^{n-1}}{\alpha_n} (H_{31}^* + H_{23}^* y^*) \tag{A17}$$



The coefficients $C_{1m}$ and $C_{2m}$ in Eq. (A16) are determined by Eq. (A5) and the result is

$$C_{2m} = -C_{1m} \tag{A18}$$

Similarly, the coefficients $D_{1n}$ and $D_{2n}$ in Eq. (A17) are determined based on Eq. (A10) as

$$D_{1n} = -D_{2n}e^{-2\Omega_{2n}} \tag{A19}$$

Substituting Eq. (A18) into Eq. (A16), the inversion of $\bar{\phi}_1^*$ leads to Eq. (12) for dimensionless hydraulic head distribution in region 1. Similarly, the inversion of $\bar{\phi}_2^*$ for region 2 after substituting Eq. (A19) into Eq. (A17) results in Eq. (13). Based on Eqs. (A10) and (A11), the coefficients of $C_{1m}$ and $D_{2n}$ can be simultaneously determined and the results are respectively given in Eqs. (18) and (19).

## Appendix B. Transient solutions for an L-shaped aquifer

Multiplying Eq. (A1) by $sin(\lambda_i x^*)$ and integrating it for $x^*$ from 0 to 1 in region 1 with Eqs. (A3) and (A4), Eq. (A1) can be transformed as

$$-\Omega_{1i}^2 \bar{\phi}_1^* - \theta_1^2 \lambda_i (-1)^i (h_{31}^* + h_{23}^* y^*) + \frac{\partial^2 \bar{\phi}_1^*}{\partial y^{*2}} = \frac{\partial \bar{\phi}_1^*}{\partial t^*} - \sum_{k=1}^M Q_{1k}^* sin(\lambda_i x_{1k}^*) \delta(y^* - y_{1k}^*) \tag{B1}$$

with

$$\bar{\phi}_1^* = \int_0^1 \phi_1^* sin(\lambda_i x_{1k}^*) \, dx^* \tag{B2}$$

where $\Omega_{1i} = \lambda_i \sqrt{\kappa_1} b_2 / l_1$, $\theta_1 = \sqrt{\kappa_1} b_2 / l_1$, and $\lambda_i = i\pi$, $i = 1,2,3, ....$

Similarly, Eq. (A2) can be transformed via multiplying Eq. (A2) by $cos(\alpha_j x^*)$ and integrating it for $x^*$ from $l_2^*$ to 1 in region 2 with Eqs. (A7) and (A8). The result is

$$-\Omega_{2j}^2 \bar{\phi}_2^* + \theta_2^2 \alpha_j (-1)^j (H_{31}^* + H_{23}^* y^*) + \frac{\partial^2 \bar{\phi}_2^*}{\partial y^{*2}} = \frac{K_{y1} S_{s2}}{K_{y2} S_{s1}} \frac{\partial \bar{\phi}_2^*}{\partial t^*} - \sum_{l=1}^N Q_{2l}^* cos(\alpha_j x_{2l}^*) \delta(y^* - y_{2l}^*) \tag{B3}$$

with

$$\bar{\phi}_2^* = \int_{l_2^*}^1 \phi_2^* cos(\alpha_j x_{2l}^*) \, dx^* \tag{B4}$$

where $\Omega_{2j} = \alpha_j \sqrt{\kappa_2} b_2 / l_1$, $\theta_2 = \sqrt{\kappa_2} b_2 / l_1$, and $\alpha_j = (1 - 1/2)\pi / (1 - l_2^*)$ for $j = 1,2,3, ....$

Then, taking Laplace transforms to Eq. (B1) results in

$$-\Omega_{1i}^2 \bar{\bar{\phi}}_1^* - \frac{1}{p} \theta_1^2 \lambda_i (-1)^i (h_{31}^* + h_{23}^* y^*) + \frac{\partial^2 \bar{\bar{\phi}}_1^*}{\partial y^{*2}} = p\bar{\bar{\phi}}_1^* - \bar{\phi}_{1s}^* - \frac{1}{p} \sum_{k=1}^M Q_{1k}^* sin(\lambda_i x_{1k}^*) \delta(y^* - y_{1k}^*) \tag{B5}$$

where $\bar{\phi}_{1s}^*$ is the steady state solution of region 1. Hence, Eq. (B5) can be organized as:



$$-\mu_i^2 + \bar{\bar{\phi}}_1^* + \frac{\partial^2 \bar{\bar{\phi}}_1^*}{\partial y^{*2}} = \frac{1}{p}\theta_1^2\lambda_i(-1)^i(h_{31}^* + h_{23}^*y^*) - \sum_{m=1}^{\infty}\Delta_1[C_{1m}E_1(m,y^*) + F_1(m,y^*)]\left(\frac{1}{2} - \frac{sin2\lambda_i}{4\lambda_i}\right) -$$

$$\frac{1}{p}\sum_{k=1}^{M}Q_{1k}^* sin(\lambda_i x_{1k}^*)\,\delta(y^* - y_{1k}^*) \tag{B6}$$

where $\mu_i = \sqrt{\theta_1^2\lambda_i^2 + p}$ with the Laplace transform of $\bar{\phi}_1^*$ defined as:

$$\bar{\bar{\phi}}_1^*(i,y^*,p) = \int_0^{\infty}\bar{\phi}_1^*(i,y^*,t)e^{-pt^*}dt^* \tag{B7}$$

Similarly, the Laplace transform of Eq. (B3) is obtained as:

$$-\Omega_{2j}^2\bar{\phi}_2^* + \frac{1}{p}\theta_2^2\alpha_j(-1)^j(H_{31}^* + H_{23}^*y^*) + \frac{\partial^2\bar{\phi}_2^*}{\partial y^{*2}} = \frac{K_{y1}S_{s2}}{K_{y2}S_{s1}}(p\bar{\bar{\phi}}_2^* - \bar{\phi}_{2s}^*) - \frac{1}{p}\sum_{l=1}^{N}Q_{2l}^* cos(\alpha_j x_{2l}^*)\,\delta(y^* - y_{2l}^*) \tag{B8}$$

where $\bar{\phi}_{2s}^*$ is the steady state solution of region 2. Thus, Eq. (B8) can be written as:

$$-\theta_j^2\bar{\bar{\phi}}_2^* + \frac{\partial^2\bar{\bar{\phi}}_2^*}{\partial y^{*2}} = -\frac{1}{p}\theta_2^2\alpha_j(-1)^j(H_{31}^* + H_{23}^*y^*) - \frac{K_{y1}S_{s2}}{K_{y2}S_{s1}}\sum_{n=1}^{\infty}\Delta_2[D_{2n}E_2(m,y^*) + F_2(m,y^*)]cos[\alpha_j(x^* - l_2^*)] -$$

$$\frac{1}{p}\sum_{l=1}^{N}Q_{2l}^* cos(\alpha_j x_{2l}^*)\,\delta(y^* - y_{2l}^*) \tag{B9}$$

where $\theta_j = \sqrt{\theta_2^2\alpha_j^2 + ps_{s2}k_{y1}/s_{s1}k_{y2}}$ with the Laplace transform of $\bar{\phi}_2^*$ defined as:

$$\bar{\bar{\phi}}_2^*(j,y^*,p) = \int_0^{\infty}\bar{\phi}_2^*(j,y^*,t)e^{-pt^*}dt^* \tag{B10}$$

The general solution of Eq. (B6) can be expressed as:

$$\bar{\bar{\phi}}_1^*(i,y^*,p) = T_{1i}e^{\mu_i y^*} + T_{2i}e^{-\mu_i y^*} + \bar{\bar{\phi}}_{1p}^*(i,y^*,p) \tag{B11}$$

where the particular solution $\bar{\bar{\phi}}_{1p}^*(i,y^*,p)$ is

$$\bar{\bar{\phi}}_{1p}^*(i,y^*,p) = \frac{e^{\mu_i y^*}}{2\mu_i}\int e^{-\mu_i y^*}\Delta_{1y}(i,y^*,p)dy^* - \frac{e^{-\mu_i y^*}}{2\mu_i}\int e^{\mu_i y^*}\Delta_{1y}(i,y^*,p)dy^* \tag{B12}$$

with

$$\Delta_{1y}(i,y^*,p) = \frac{1}{p}\theta_1^2\lambda_i(-1)^i(h_{31}^* + h_{23}^*y^*) - \sum_{m=1}^{\infty}\Delta_1[C_{1m}E_1(m,y^*) + F_1(m,y^*)]\left(\frac{1}{2} - \frac{sin2\lambda_i}{4\lambda_i}\right) -$$

$$\frac{1}{p}\sum_{k=1}^{M}Q_{1k}^* sin(\lambda_i x_{1k}^*)\,\delta(y^* - y_{1k}^*) \tag{B13}$$

Moreover, Eq. (B9) can also be expressed as:

$$\bar{\bar{\phi}}_2^*(j,y^*,p) = T_{1j}e^{\theta_j y^*} + T_{2j}e^{-\theta_j y^*} + \bar{\bar{\phi}}_{2p}^*(j,y^*,p) \tag{B14}$$

in which $\bar{\bar{\phi}}_{2p}^*(j,y^*,p)$ is





$$\bar{\bar{\phi}}_{2p}^*(j, y^*, p) = \frac{e^{\theta_j y^*}}{2\theta_j} \int e^{-\theta_j y^*} \Delta_{2y}(j, y^*, p) dy^* - \frac{e^{-\theta_j y^*}}{2\theta_j} \int e^{\theta_j y^*} \Delta_{2y}(i, y^*, p) dy^* \tag{B15}$$

with

$$\Delta_{2y}(j, y^*, p) = -\frac{1}{p}\theta_2^2\alpha_j(-1)^j(H_{31}^* + H_{23}^* y^*) - \frac{K_{y1}S_{s2}}{K_{y2}S_{s1}}\sum_{n=1}^{\infty}\Delta_2[D_{2n}E_2(m, y^*) + F_2(m, y^*)]\cos[\alpha_j(x^* - l_2^*)] -$$

$$\frac{1}{p}\sum_{l=1}^{N}Q_{2l}^*\cos(\alpha_j x_{2l}^*)\,\delta(y^* - y_{2l}^*) \tag{B16}$$

On the basis of Eq. (A5), the coefficient $T_{2i}$ in Eq. (B11) can be determined in terms of $T_{1i}$ as:

$$T_{2i} = -h_{31}^* \frac{(-1)^i}{\lambda_i} + \frac{h_{31}^*}{p\mu_i^2}[\theta_1^2\lambda_i(-1)^i] + \Delta_1(-1)^i C_{1m}\left(\frac{1}{2} - \frac{sin2\lambda_i}{4\lambda_i}\right) - T_{1i} \tag{B17}$$

The solution for hydraulic head distribution in region 1 is given as Eq. (26) which is obtained by substituting Eq. (B17) into Eq. (B11) and then taking the following inverse Fourier transform to Eq. (B11) denoted as:

$$\tilde{\phi}_1^*(x^*, y^*, p) = \sum_{i=0}^{\infty}\bar{\phi}_1^*(i, y^*, p)sin(\lambda_i x^*) \tag{B18}$$

with

$$w_{1i}^* = T_{1i}(e^{\mu_i y^*} - 1)\big|_{y^*=b_1^*} - \frac{1}{2\mu_i p}\sum_{k=1}^{M}Q_{1k}^*sin(\lambda_i x_{1k}^*)\left[e^{\mu_i(y^*-y_{1k}^*)} - e^{\mu_i(y_{1k}^*-y^*)}\right]\big|_{y^*=b_1^*} \tag{B19}$$

Similarly, $T_{1j}$ in Eq. (B14) can be obtained based on Eq. (A9) as:

$$T_{1j} = T_{2j}e^{-2\theta_j} - \left[\frac{\theta_2^2\alpha_j(-1)^j}{p} + \frac{K_{y1}S_{s2}(-1)^j}{K_{y2}S_{s1}\alpha_j}\right]\frac{H_{21}^*e^{-\theta_j}}{\theta_j^2} + \frac{H_{21}^*e^{-\theta_j}(-1)^j}{\alpha_j p} + \frac{1}{p}\sum_{l=1}^{N}Q_{2l}^*\cos(\alpha_j x_{2l}^*)\frac{e^{-\theta_j y_{2l}^*} - e^{\theta_j(y_{2l}^*-2)}}{2\theta_j} \tag{B20}$$

The solution for region 2 is Eq. (27) which is acquired by substituting Eq. (B20) into Eq. (B14) then taking the following inverse Fourier transform to Eq. (B14) expressed as:

$$\tilde{\phi}_2^*(x^*, y^*, p) = \sum_{j=0}^{\infty}\bar{\phi}_2^*(j, y^*, p)cos(\alpha_j x^*) \tag{B21}$$

with

$$w_{2j}^* = T_{2j} \tag{B22}$$

Furthermore, the coefficients of $w_{1i}^*$ and $w_{2j}^*$ can be simultaneously determined by Eqs. (A10) and (A11). The results are respectively given in Eqs. (36) and (37).



**Table 1 Notations used in the text.**

| Notation | Definition |
| --- | --- |
| $\phi_1,\ \phi_2$ | Hydraulic head for region 1 and 2. [L] |
| $Q_{1k},\ Q_{2k}$ | Unit thickness pumping rate for region 1 and 2. [$L^2$/T] |
| $S_{s1},\ S_{s2}$ | Specific storage for region 1 and 2. [$L^{-1}$] |
| $K_x,\ K_y$ | Hydraulic conductivities in x- and y-direction. [L/T] |
| $t$ | Time. [L] |
| $p$ | Laplace variable. |
| $h_1,\ h_2,\ h_3$ | Hydraulic heads at boundaries AG, DE and point B, respectively. [L] |
| $l_1,\ l_2$ | Length of boundary FG and AB. [L] |
| $b_1,\ b_2$ | Length of boundary BC and CD. [L] |
| $\kappa_1,\ \kappa_2$ | Anisotropic ratio of hydraulic conductivity in region 1 and 2. |
| $\Delta_1$ | $\begin{cases} 1, & m = 0 \\ 2, & m \neq 0 \end{cases}, m = 1,2,3, \ldots.$ |
| $\Delta_2$ | $\frac{2}{1-l_2^*}.$ |
| $\lambda_v$ | $\frac{v\pi}{l_1^*}, v = m, i = 1,2,3, \ldots.$ |
| $\alpha_w$ | $\frac{(w-1/2)\pi}{1-l_2^*}, w = n, j = 1,2,3, \ldots.$ |
| $\Omega_{1v}$ | $\lambda_v \sqrt{\kappa_1}\, b_2/l_1, v = m, i = 1,2,3, \ldots.$ |
| $\Omega_{2w}$ | $\alpha_w \sqrt{\kappa_2}\, b_2/l_1, w = n, j = 1,2,3, \ldots.$ |
| $h_{21}{}^*$ | $(h_2 - h_1)/h_1$ |
| $h_{23}{}^*$ | $(h_2 - h_3)/h_1$ |
| $h_{31}{}^*$ | $(h_3 - h_1)/h_1$ |
| $H_{21}{}^*$ | $(h_2 - h_1)/h_2$ |
| $H_{23}{}^*$ | $(h_2 - h_3)/h_2$ |
| $H_{31}{}^*$ | $(h_3 - h_1)/h_2$ |





| | |
|---|---|
| $\delta_1$ | $2$ |
| $\delta_2$ | $\dfrac{2}{1 - l_2^*}$ |
| $\theta_1$ | $\sqrt{\kappa_1 (b_2^2 / l_1^2)}$ |
| $\theta_2$ | $\sqrt{\kappa_2 (b_2^2 / l_1^2)}$ |
| $\mu_i$ | $\sqrt{\theta_1^2 \lambda_i^2 + p}, i = 1,2,3, \dots$ |
| $\theta_j$ | $\sqrt{\theta_2^2 \alpha_j^2 + p s_{s2} k_{y1} / s_{s1} k_{y2}}, j = 1,2,3, \dots$ |



**Figures**

**Figure 1: Location of the fluvial aquifer. Note that this figure is modified from Google Earth.**





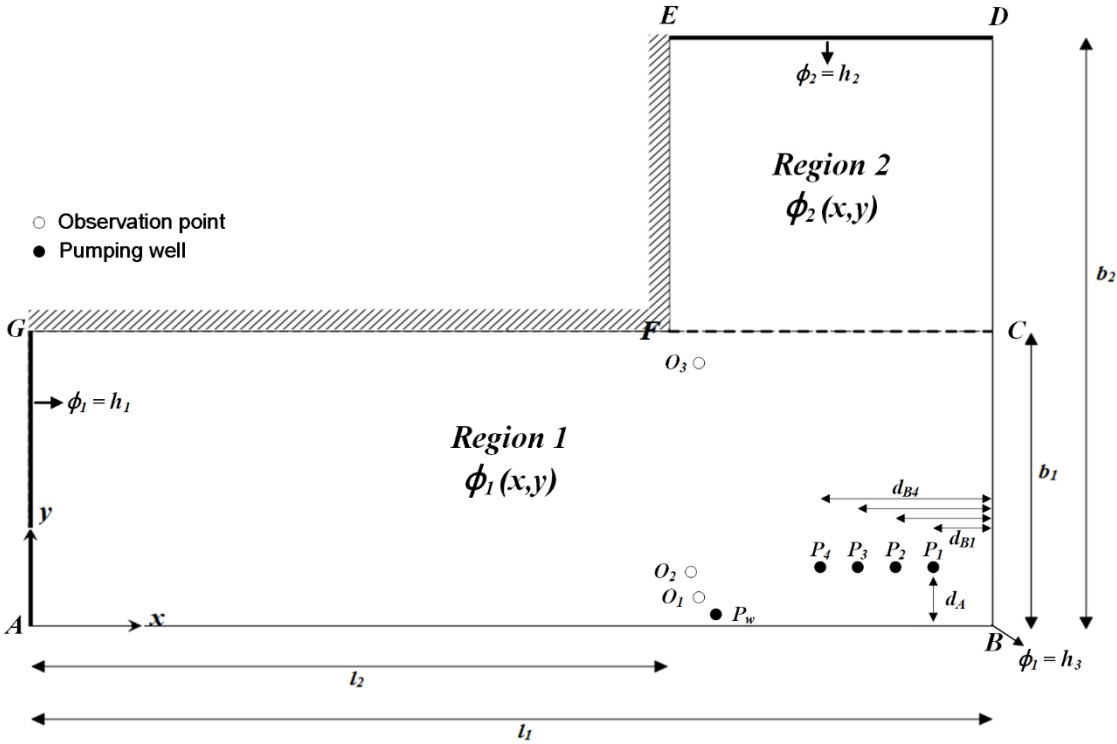

**Figure 2: The L-Shaped fluvial aquifer with two sub-regions.**



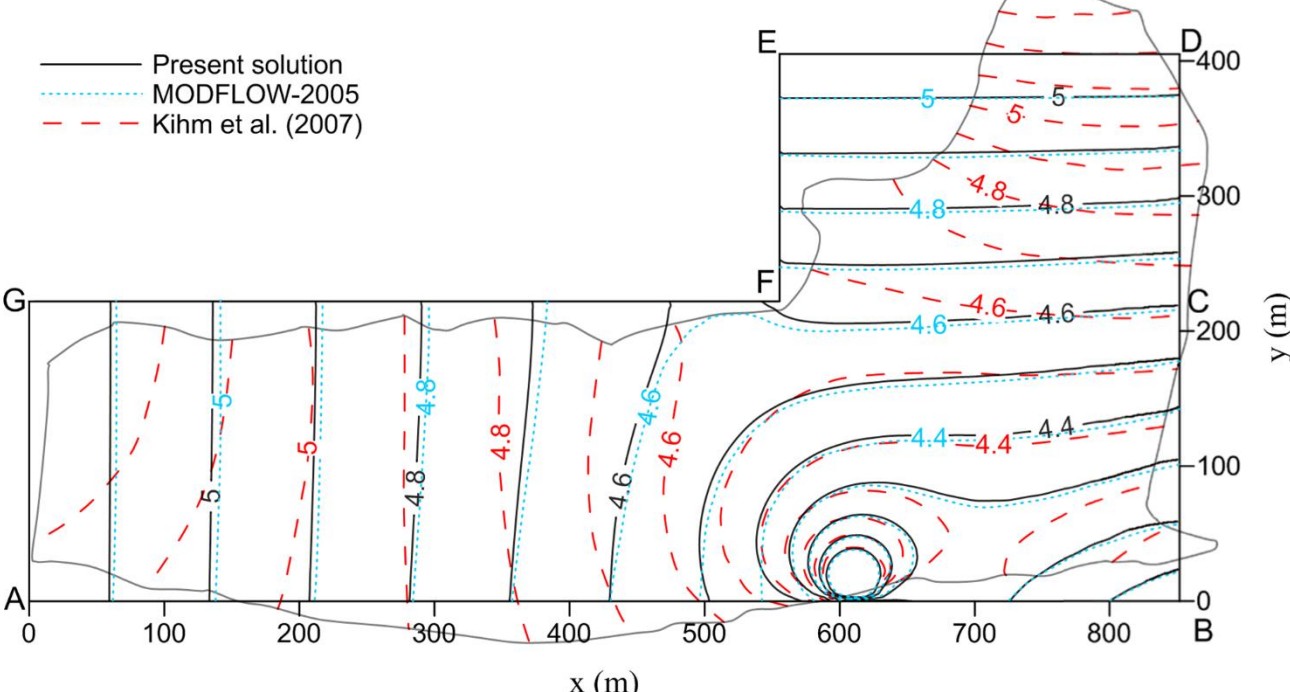

**Figure 3: Contours of hydraulic head in L-shaped aquifer predicted by the present solution, MODFLOW-2005, and FEM simulations with irregular outer boundary reported in Kihm et al. (2007).**





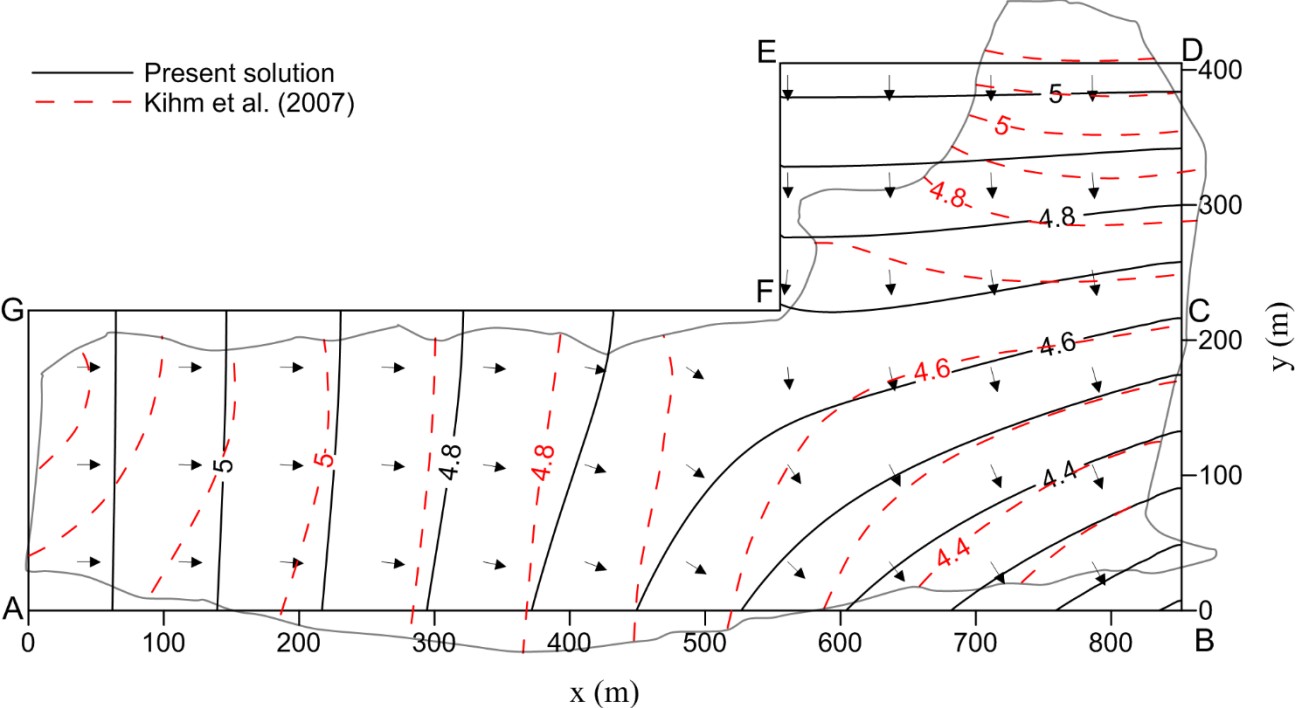

**Figure 4: Steady-state hydraulic head contours without pumping in Yongpoong 2 Agriculture District.**





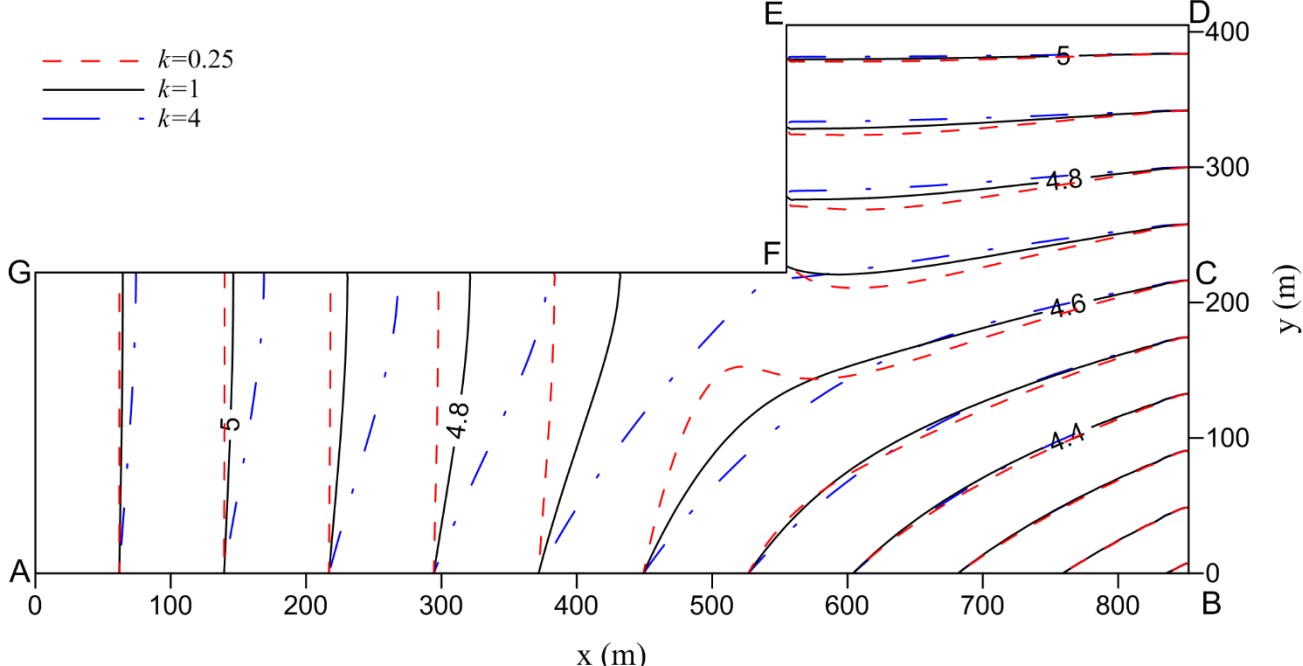

**Figure 5: Steady-state hydraulic head contours in the L-shaped aquifers with three different anisotropy ratios for $\kappa_1 = \kappa_2 = \kappa$.**





**Figure 6: Temporal hydraulic head distributions observed at piezometers $O_1$, $O_2$, and $O_3$. In the subscript, M, F, and A represents field measurements, FEM simulations, and predictions from present solution, respectively.**

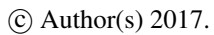





**Figure 7: Temporal distributions of *SDR*s and *SRR* due to pumping at $P_w$.**