# Peer review of "Analysis of groundwater flow and stream depletion in the L-shaped fluvial aquifer"

_Hydrology and Earth System Sciences, 2017_

## Referee Comment (RC1) · Anonymous Referee #1 · 12 Oct 2017

1- General comments There are few analytical studies addressing flow field in multi-aquifer systems. This may be attributed to inadequacy of conventional solution techniques in dealing with such geometrically cornered entities. Aimed at reproducing a real-world scenario in a semi-analytical framework, the present manuscript also offers useful insights regarding the nature of multi-well hydraulics in L-shaped aquifers consisting of two anisotropic sub-regions with properly imposed interface conditions. Comparisons are also made with relevant numerical results and existing measurement data. The subject can further be clarified if the authors consider the comments listed below:

2-Specific comments

2-1 In addition to those reviewed in "Introduction", the following studies examine different ways of simplifying natural aquifer settings through non-rectangular domains: Variational method of Kantorovich for modeling rainfall induced mounds in trapezoidal-shaped aquifers (Mahdavi and Seyyedian, 2014); the method of Strack's discharge potential for groundwater hydraulics in coastal promontories (Kacimov et al., 2016); and more recently, holomorphic functions for flow fields defined in circular meniscus (Kacimov et al., 2017). Moreover, the case of L-shaped domains has been treated analytically in different fields of engineering such as torsion of elastic bars (Kantorovich and Krylov, 1958) as well as heat conduction in plates (Mackowski, 2011). It is suggested to include above-mentioned works in the literature review.

2-2 Since (46) refers to water exchange along aquifer-stream interface AB (denoted by ), it should take into account only contribution from hydraulic gradients in Region 1, i.e. the portion of aquifer which is directly in hydraulic connection with the stream. The second integral in this expression, which implies direct influence of Region 2 on SDR_A, thus seems irrelevant and should be removed. 2-3 When evaluating SDR_B, the first and second integrals in (47) should be taken from 0 to b1 and from b1 to b2, respectively, for the same reasoning as described before. 2-4 The extraction water comes from surrounding streams and compression of fully-saturated porous media, as clearly mentioned in the manuscript. Contribution from constant-head boundaries (AG and ED) is, however, ignored in the aquifer water-budget model and only the effects of AB and BD are addressed by (50). Obviously, Darcian flow (either inwardly or outwardly) is induced by non-zero head gradients perpendicular to AG and ED. Such water fluxes are also disregarded in Fig. 7.

3-Technical corrections 3-1 The dimension of 1D Dirac's delta function should be mentioned: [1/L] 3-2 The dimension of time should be changed to [T] in "Table 1". 3-3 Unbalanced parenthesis is detected in (34). 3-4 Equal sign is omitted in (24) and (25).

References:

Kacimov, A. R., Kayumov, I. R., Al-Maktoumi, A., 2016. Rainfall induced groundwater mound in wedge-shaped promontories: The Strack–Chernyshov model revisited. Advances in Water Resources, 97, 110–119.

Kacimov, A. R., Maklakov, D. V., Kayumov, I. R., Al-Futaisi, A., 2017. Free Surface flow in a microfluidic corner and in an unconfined aquifer with accretion: The Signorini and Saint-Venant analytical techniques revisited. Transport in Porous Media, 116(1), 115–142.

Kantorovich, L.V., Krylov, V.I., 1958. Approximate Methods of Higher Analysis. Interscience, New York.

Mackowski, D. W. (2011). Conduction Heat Transfer: Notes for MECH 7210. Mechanical Engineering Department, Auburn University.

Mahdavi, A., Seyyedian, H., 2014. Steady-state groundwater recharge in trapezoidal-shaped aquifers: A semi-analytical approach based on variational calculus. Journal of Hydrology, 512, 457–462.

---

## Referee Comment (RC2) · Anonymous Referee #2 · 29 Oct 2017

The paper provides an analytical solution for transient groundwater flow in an L-shaped aquifer, with strong connection to a stream. The so called analytical solution is not completely analytical, as numerical tools as the Stehfest algorithm are included to obtain the final result. When the results are compared with a MODFLOW solution, in fact two quite different numerical approaches are compared. Both of these approaches have their limitations and deliver approximate solutions only. The possible size of the errors is difficult to discuss and is not addressed in the manuscript.

Usually analytical solutions are utilized for benchmarking numerical codes, because they are a more accurate representation of the exact solution. Obviously this property is not expected by the authors, when they present their approach. In contrary they use a numerical solution for benchmarking their method, not taking into account that the

numerical solution is definitely only an approximation.

Concerning the model region, the L-shaped domain is surely a big deviation form the real aquifer geometry, especially along boundary AG, but even more along boundaries FE and ED. Thus deviances, as shown in Fig. 3 could be expected. The problem with the manuscript is that it cannot trace back the differences to its causes: it could be the different numerical approach (MODFLOW, FEM, 'analytical') or the different model region. Were the results of the numerical models obtained with sufficient mesh refinement?

The production well is located quite near to the boundary AB. It can be expected that the strong head gradients that appear due to this constellation can only be reproduced numerically if strong mesh refinement is used in the direct vicinity of the well. Concerning the real world situation, it could be doubted that a numerical approach with a constant head boundary can address the physically relevant processes in that case. I would expect that strong or weak connection between aquifer and surface water body play a role in reality in addition.

If the paper could be re-written in a way to address the points made, I could deliver a more positive comment.

---

## Short Comment (SC1) · 15 Dec 2017

The paper under review presents a semi-analytic method for describing groundwater flow in an irregular (L-shaped) unconfined aquifer bounded on two sides by contributing streams. The authors have presented a solution for groundwater flow in a steady-state condition, and, using the steady-state solution as a boundary condition, under the influence of a single pumping well. The authors' work is developed from the work of Kihm et al. in the 2007 paper "Three-dimensional numerical simulation of fully coupled groundwater flow and land deformation due to groundwater pumping in an unsaturated fluvial aquifer system" and draws heavily from the conceptual model developed therein.

Substantive Praise-Worthy Aspects:

[Figure]

In this paper, the authors present a novel method for solving for the groundwater flow field for a complex hydrogeology problem. As noted by anonymous referee #1, few papers address groundwater flow in multi-unit aquifers with complex shape, so by presenting a semi-analytic solution to groundwater flow under these conditions, this paper provides insight into methodology for representing hydrologic processes.

The problem addressed by the authors also provides insight into modelling the relative contribution of aquifer storage and stream filtration water to the total water abstracted from a pumping well. The authors' work also contributes to an understanding and awareness of the interaction of surface hydrology and groundwater, a topic that should be further addressed and developed. By developing a solution for the groundwater flow field in a system incorporating these factors, the authors have made a worthy contribution to the field of hydrology and engineering.

Substantive Considerations:

This paper draws heavily on the work of Kihm et al. (2007), and I am concerned that not all the material presented has been cited correctly. Several examples of incorrectly cited material are provided below:

The sentence on P.4, L.9-10 is cited as a summary, but the wording may not be sufficiently different from the original sentence in Kihm et al. (2007, P.4).

A direct quotation from Kihm et al. (2007, P.4) that was not properly indicated or cited was detected on P.4, L.17-18.

Figure 6 is an updated reprint of Kihm et al.'s Figure 12 (2007, P.12), but is not directly cited in the figure caption.

Important assumptions made in the development of the conceptual model have not been discussed. These assumptions follow those made in Kihm et al. (2007) and include the assumption that hydrostatic conditions exist in the vertical profile through both units of the aquifer (i.e. the peizometric surface is equal to the water table at all points

along the vertical profile) and that recharge to the system from vertical percolation or precipitation is negligible. These assumptions, and others, may represent significant deviations from real-world conditions, and should be explicitly stated.

Although the piezometer data presented in Kihm et al. 2007 appears to support the modelled solutions, it should be noted that piezometer observations are only available over a period of 5 days; no information is presented to validate the modelled response to pumping beyond this period.

Considering Figure 6, the 5-day observation period appears insufficient to observe any response to pumping at piezometer 3 (O3). This indicates that these data are irrelevant for the purpose of validating the transient solution for hydraulic head distribution presented by the authors.

It is also to be considered that at a time period of less than 5 days, the majority of the modelled contribution of abstracted water is from aquifer storage (SRR), with the contribution from stream filtration (SDR) increasing after this point. The absence of observed piezometer response to pumping after a time of 5 days would seem to prevent any conclusions from being drawn as to the application of the method presented by the authors in predicting aquifer response to pumping in situations with a large stream filtration component.

It is my opinion that the results presented in this paper are insufficient to draw conclusions as to the validity of the methodology presented in predicting aquifer response to pumping. The results presented, however, demonstrate consistency between the semi-analytic method presented by the authors and the numeric model developed by Kihm et al. for the same aquifer system. Likewise, it is my opinion that the results presented are insufficient to draw conclusions regarding the significance of unsaturated flow and land deformation due to the limited observed data.

The authors present the semi-analytic solution as a design tool for determining well location. The demonstrated applicability of the numeric simulations presented by Kihm

et al. 2007 and the authors' solution developed in MODFLOW, validated by the semi-analytic method presented in this paper, would seem to be more flexible and appropriate tools for the design of well location.

Further to the substantive observations which I have made above, there are several additional observations of a less critical nature that I would like to make.

Strengths:

The derivation of the analytic solution appears well documented and described. This paper provides the reader with a clear description of the analytical methods used by the authors, theoretically allowing for the results to be reproduced. The literature review presented by the authors also appears to be detailed, and well-structured, providing valuable information to other scientists interested in studying groundwater flow in aquifers with complex boundaries and that are bounded by contributing streams.

Areas of Improvement:

The assumption was made that all flow is horizontal, including the flow through the overlying clay loam aquitard unit, which has been assigned a hydraulic conductivity two orders of magnitude lower than the underlying loamy sand unit. This assumption is necessary for the simplification of the groundwater flow equation to 2-dimensions, but is non-realistic and the implications of this assumption have not been addressed by the authors.

The equivalent hydraulic conductivity for horizontal flow (Eq. (48)), discussed on P.10, L.3-4, is calculated as the weighted arithmetic mean of the two units assuming the full thickness (2.5 m) of the overlying unit is available for groundwater flow. Since the overlying unit is only saturated to a maximum seasonal average thickness of 0.79 m (as described on P.4, L.9), it may be more appropriate to use the saturated thickness of the upper layer when calculating the equivalent hydraulic conductivity for the aquifer.

The logic regarding the required well setback from a stream is incomplete (P.12, L.7-9),

and the connection between the required well setback distance from possible contaminants and the well setback distance from a stream is not clear.

As noted by anonymous referee #2, the solution presented by the authors is semi-analytical. The first use of the term "semi-analytical" by the authors is in the conclusion on P.12, L.25. The solution presented by the authors should be consistently described throughout the paper, as appropriate.

Several minor grammatical issues were found within the paper, and are listed as follows:

P.2, L.9 typo: "arbitrarily", should be "arbitrary"

P.2, L.12-15 ambiguous references; it is not clear that the authors are referring to the work of Kihm et al. (2017)

P.2, L.12 inconsistent hyphenation of "L-shape"

P.2, L.15 missing "the": "in <the> transient case"

P.2, L. 20 poor grammar: "to solve a regional groundwater in an. . ."

P.3, L.31 "perennial stream<s>. . ."

P.4, L.22 syntax error: "Consider that there are totally M pumping wells. . ."

(Throughout paper) inconsistent use of italics to denote units, and spaces between values and units (i.e. 6m, 6 m, 6 m)

P.2, L.34 ambiguous reference to "irregular boundaries" – what are irregular boundaries?

P.3, L.18 ambiguous reference: "principle direction aligned with the border of the sub-region"; which border?

P.13, L.6-7. Awkward transition. This should either be a new paragraph, or these sentences should be rewritten.

[Figure]

Suggestions:

A careful and detailed review of the entire paper should be conducted by the authors to ensure all material is appropriately cited. The authors should revisit the description of the conceptual model and either further develop and detail the assumptions made in the development of the conceptual model or clearly state that the conceptual model and assumptions have been taken from the work of Kihm et al. (2007) and refer the readers to that paper for details.

The authors should address the implications of the simplifying assumptions when applying the results of the semi-analytic and numeric solutions for groundwater flow in this aquifer to the real world. The limitations of the 5-day observation period should be noted by the authors.

The conclusions drawn by the authors should be reconsidered. The results appear to demonstrate consistency between the semi-analytical method presented by the author and the numeric model presented by Kihm et al. (2007), and raise questions as to the significance of unsaturated flow and land deformation. Conclusions regarding real-world aquifer response to groundwater abstraction appear unsupported. It is recommended that the authors reframe their work as a method of validating the numeric simulations and as a method of developing a better understanding of the physical processes governing groundwater flow.

Reviewer Experience:

It should be noted that I am a Master of Science candidate in the field of engineering, with minimal experience in either analytical or numerical methods for describing groundwater flow. I have no prior experience refereeing academic submissions. The observations and opinions I have expressed herein should be considered with my inexperience in mind.

Proposed Fate:

[Figure]

The authors are to be commended for their approach to this complex problem. It is my opinion that the methods and results presented by Lin, Chang, and Yeh makes a valuable contribution to the field of hydrology and engineering and are of interest to the scientific community. However, the issues noted above are significant. I recommend that this paper be resubmitted for review following the revisions suggested above. I would further recommend that extreme caution be exercised by both the authors and by the editor in vetting the submission for incorrectly cited material.

References:

Kihm J H, Kim J M, Song S H, Lee G S. 2007. Three-dimensional numerical simulation of fully coupled groundwater flow and land deformation due to ground-water pumping in an unsaturated fluvial aquifer system. J. Hydrol. 335:1–14. doi:https://doi.org/10.1016/j.jhydrol.2006.09.031.

---

## Author Comment (AC1) · 9 Jan 2018

**Reply to Anonymous Referee #1**

General comments:

● There are few analytical studies addressing flow field in multiaquifer systems. This may be attributed to inadequacy of conventional solution techniques in dealing with such geometrically cornered entities. Aimed at reproducing a real-world scenario in a semi-analytical framework, the present manuscript also offers useful insights regarding the nature of multi-well hydraulics in L-shaped aquifers consisting of two anisotropic sub-regions with properly imposed interface conditions. Comparisons are also made with relevant numerical results and existing measurement data. The subject can further be clarified if the authors consider the comments listed below:

Response:

Thanks, we provide point-by-point response to each of your comment listed below. The page and line numbers given in our responses are referred to those in the revised manuscript.

Specific comments

● In addition to those reviewed in "Introduction", the following studies examine different ways of simplifying natural aquifer settings through non-rectangular domains: Variational method of Kantorovich for modeling rainfall induced mounds in trapezoidal-shaped aquifers (Mahdavi and Seyyedian, 2014); the method of Strack's discharge potential for groundwater hydraulics in coastal promontories (Kacimov et al., 2016); and more recently, holomorphic functions for flow fields defined in circular meniscus (Kacimov et al., 2017). Moreover, the case of L-shaped domains has been treated analytically in different fields of engineering such as torsion of elastic bars (Kantorovich and Krylov, 1958) as well as heat conduction in plates (Mackowski, 2011). It is suggested to include above-mentioned works in the literature review.

Response:

Thanks for the suggestion. These articles have been reviewed and listed in the revised manuscript for two parts. The first part is from lines 24-29, page 1 to lines 1-4, page 2 as: "Many studies have been devoted to the development of analytical models for describing flow in finite aquifers with a rectangular boundary …, a wedge-shaped boundary (Chan et al., 1978; Falade, 1982; Holzbecher, 2005; Yeh et al., 2008; Chen et al., 2009; Samani and Zarei-Doudeji, 2012; Samani and Sedghi, 2015; Kacimov et al. 2016), a triangle boundary (Asadi-Aghbolaghi et al., 2010) a trapezoidal-shaped boundary (Mahdavi and Seyyedian, 2014), or a

meniscus-shaped domain (Kacimov et al. 2017). So far, the case of re-entrant angle (L-shaped) boundaries has been treated analytically in different fields such as torsion of elastic bars (Kantorovich and Krylov, 1958), head fluctuation problems for tidal aquifers (Sun, 1997; Li and Jiao, 2002), and heat conduction in plates (Mackowski, 2011). However, none of them are to deal with pumping or stream depletion problems."

Then, the second part of the new reviews is given after the sentence "Patel and Serrano (2011) solved nonlinear boundary value problems of multidimensional equations by Adomian's method of decomposition for groundwater flow in irregularly shaped aquifer domains." in lines 11-19, page 3 as "Mahdavi and Seyyedian (2014) developed a semi-analytical solution for hydraulic head distribution in trapezoidal-shaped aquifers in response to diffusive recharge of constant rate. The aquifer was surrounded by four fully penetrating and constant-head streams. Kacimov et al. (2016) used the Strack-Chernyshov model to investigate the unconfined groundwater flows in a wedge-shaped promontories with accretion along the water table and outflow from a groundwater mound into draining rays. Huang et al. (2016) presented 3D analytical solutions for hydraulic head distributions and *SDR*s induced by a radial collector well in a rectangular confined or unconfined aquifer bounded by two parallel streams and no-flow boundaries. Currently, the distribution of groundwater flow velocity in a circular meniscus aquifer was investigated analytically by theory of holomorphic functions and numerically by FEM (Kacimov et al., 2016)."

- Since (46) refers to water exchange along aquifer-stream interface AB (denoted by), it should take into account only contribution from hydraulic gradients in Region 1, i.e. the portion of aquifer which is directly in hydraulic connection with the stream. The second integral in this expression, which implies direct influence of Region 2 on SDR$_A$, thus seems irrelevant and should be removed. When evaluating SDR$_B$, the first and second integrals in (47) should be taken from 0 to b1 and from b1 to b2, respectively, for the same reasoning as described before.

Response:

Thanks for the comment. The stream depletion rates (in Laplace domain) from stream reaches AB and BD have been modified, respectively, as

$$\widetilde{SDR}_A = \frac{q_A}{Q} = -\frac{1}{Q} \int_0^{l_1} K_{y1} \left. \frac{\partial \tilde{\phi}_1(x,y,p)}{\partial y} \right|_{y=0} dx \qquad (A1)$$

and

$$\widetilde{SDR}_B = \frac{q_B}{Q} = \frac{1}{Q} \left( \int_0^{b_1} K_{x1} \left. \frac{\partial \tilde{\phi}_1(x,y,p)}{\partial x} \right|_{x=l_1} dy + \int_{b_1}^{b_2} K_{x2} \left. \frac{\partial \tilde{\phi}_2(x,y,p)}{\partial x} \right|_{x=l_1} dy \right) \qquad (A2)$$

in equations (46) and (47) in the revised manuscript.

- The extraction water comes from surrounding streams and compression of fully-saturated porous media, as clearly mentioned in the manuscript. Contribution from constant-head boundaries (AG and ED) is, however, ignored in the aquifer water-budget model and only the effects of AB and BD are addressed by (50). Obviously, Darcian flow (either inwardly or outwardly) is induced by non-zero head gradients perpendicular to AG and ED. Such water fluxes are also disregarded in Fig. 7.

Response:

Thanks for the comment. We replace the sentence "The hydraulic heads along AG and DE are fixed at their average water stages as did in Kihm et al. (2007)." with the following text "The hydraulic heads along AG and DE are assumed equal to their average head values as did in Kihm et al. (2007). In other words, the boundaries along AG and ED are assumed under the constant-head condition in our mathematical model. Physically, they are not streams and therefore not count for their contribution in the calculations of SDR in Sect. 2.5 Stream depletion rate." (lines 17-20, page 4 in the revised manuscript). Note that we also evaluate the SDRs along the boundaries AG and ED and their estimated values are both less than 0.0008 over the entire pumping period, indicating that their effects are negligible.

Technical corrections

- The dimension of 1D Dirac's delta function should be mentioned: [1/L]

Response:

Thanks, it has been added as: "The symbol $\delta$ represents one dimensional (1D) Dirac's delta function $[1/T]$." (line 12, page 5)

- The dimension of time should be changed to [T] in "Table 1".

Response:

Thanks, it has been corrected.

- Unbalanced parenthesis is detected in (34).

Response:

Done as suggested.

- Equal sign is omitted in (24) and (25).

Response:

Thanks, it has been corrected.

References:

Kacimov, A. R., Kayumov, I. R., and Al-Maktoumi, A.: Rainfall induced groundwater mound in wedge-shaped promontories: The Strack–Chernyshov model revisited. Advances in Water Resources, 97, 110–119, 2016.

Kacimov, A. R., Maklakov, D. V., Kayumov, I. R., and Al-Futaisi, A.: Free Surface flow in a microfluidic corner and in an unconfined aquifer with accretion: The Signorini and Saint-Venant analytical techniques revisited. Transport in Porous Media, 116(1), 115–142, 2017

Kantorovich, L.V., and Krylov, V.I.: Approximate Methods of Higher Analysis. Interscience, New York, 1958.

Mackowski, D. W.: Conduction Heat Transfer: Notes for MECH 7210. Mechanical Engineering Department, Auburn University, 2011.

Mahdavi, A., and Seyyedian, H.: Steady-state groundwater recharge in trapezoidal-shaped aquifers: A semi-analytical approach based on variational calculus. Journal of Hydrology, 512, 457–462, 2014

[Figure]

**Modified Figure 7. Temporal distributions of $SDR$s, $CHR$s and $SRR$ due to pumping at $P_w$.**

---

## Author Comment (AC2) · 9 Jan 2018

**Reply to the comments of Referee #2**

(Note that the page number and line number mentioned in the following responses are referred to those in the revised manuscript.)

● The paper provides an analytical solution for transient groundwater flow in an L-shaped aquifer, with strong connection to a stream. The so called analytical solution is not completely analytical, as numerical tools as the Stehfest algorithm are included to obtain the final result. When the results are compared with a MODFLOW solution, in fact two quite different numerical approaches are compared. Both of these approaches have their limitations and deliver approximate solutions only. The possible size of the errors is difficult to discuss and is not addressed in the manuscript.

Response:

1. Thanks for reviewer's reminder. The steady state solution derived in this study is analytical and the transient solution is semi-analytical because it needs a numerical tool to obtain the time-domain result. To avoid confusion, we therefore use the word "semi-analytical" in lieu of "transient" before the time-domain solution in the revised manuscript.

2. Figure A (Figure 3 in the revised manuscript) depicts the hydraulic head contours in L-shaped aquifer simulated by the present solution and MODFLOW. As shown in this figure, the head distribution simulated by the present solution agrees with that by MODFLOW except in the region near the no-flow boundary FG, which has the largest relative deviation 2.1% between these two models. Furthermore, field observations are available from Kihm et al. (2007) to compare the simulation results from the present solution and MODFLOW. Figure B (Figure 6 in the revised manuscript) plots the temporal hydraulic head distribution obtained from the present solution, MODFLOW, and FEM from Kihm et al. (2007) at piezometers $O_1$, $O_2$ and $O_3$, together with the field observations at these piezometers. Compare to the field data, the largest deviation is 0.03m and 0.08m for both MODFLOW and present solution at $O_3$ and $O_2$, respectively, and 0.04m for MODFLOW and 0.07m for present solution at $O_1$. The discussion of the comparison is addressed in the revised manuscript in lines 26-29, page 10 as "The hydraulic head distribution predicted by the present solution of Eqs. (26) and (27) and represented by the dotted line is shown in Figure 3. The figure indicates that the head distribution simulated by the present solution agrees with that by MODFLOW except in the region near the no-flow boundary FG, which has the largest relative

deviation 2.1% between these two models. The comparison of the head distributions predicted by the present solution and MODFLOW ensures that the simplification of aquifer layers in the present model is appropriate and gives a fairly good predicted results." and lines 6-11, page 12 as "Compared with the field observation, the differences of predicted hydraulic head among FEM, present solution and MODFLOW are all less than 0.08m at these three piezometers during 0.1 to 10 day. In addition, the largest relative differences between measured heads and predicted heads by the present solution at $O_1$ to $O_3$ are respectively 1.64%, 1.74% and 0.62%, indicating that the present solution gives good predictions in the early pumping period. Moreover, the effects of unsaturated flow and land deformation on the groundwater flow in Yongpoong aquifer are small and may be negligible."

- Usually analytical solutions are utilized for benchmarking numerical codes, because they are a more accurate representation of the exact solution. Obviously this property is not expected by the authors, when they present their approach. In contrary they use a numerical solution for benchmarking their method, not taking into account that the numerical solution is definitely only an approximation.

Response:

We agree that analytical solutions are the primary means for benchmarking numerical codes. Here we would like to mention that the use of MODFLOW is to examine the suitability of simplification made in our analytical solution using the approach of equivalent hydraulic conductivity. To avoid confusion, the title of section 3 "Solution validation and application" and subsection 3.1 "Solution validation by MODFLOW-2005" in page 9 in original manuscript is respectively replaced by the "Comparisons of present solution, numerical solutions and field observed data" and "Comparisons of present solution with MODFLOW solution". The purpose of the MODFLOW simulation is further discussed in lines 3-31, page 10 in the revised manuscript with the following text: "The software MODFLOW is used to simulate the groundwater flow due to pumping in the L-shaped aquifer in Yongpoong 2 Agriculture District with different hydraulic conductivities for the two layers. The MODFLOW is a widely used finite-difference model developed by U.S. Geological Survey for the simulation of 3D groundwater flow problems under various hydrogeological conditions (USGS, 2005). As shown in Figure 1, region 1 has an area of $852m \times 222m$ (i.e., $l_1 \times d_1$) while the area of region 2 is $297m \times 183m$ (i.e., $(l_1 - l_2) \times (d_2 - d_1)$). Thus, the total area of these two regions is $243495 \ m^2$ which is close to the area of the fluvial aquifer ($246500m^2$) reported in Kihm et al. (2007). In the simulation of MODFLOW,

the plane of the L-shaped aquifer is discretized with a uniform cell size of $3m \times 3m$. The aquifer thickness is $6m$ and divided into two layers. The upper loam layer is $2.5m$ and lower sand layer $3.5m$. Within the aquifer domain, there is totally 54110 cells while the numbers of cell are 42032 and 12078 respectively for region 1 and region 2. The types of outer boundary specified for the L-shaped aquifer are the same as those defined in the mathematical model. The hydraulic heads along AG and DE are respectively $h_1 = 5.18m$ and $h_2 = 5.29m$ and the head at point B is $h_3 = 4.06m$. The fluvial aquifer reported in Kihm et al. (2007) is isotropic and homogeneous in horizontal direction. In other words, the hydraulic conductivities in $x$ and $y$ directions are identical in both regions 1 and 2 (i.e., $K_{x1} = K_{y1} = K_{x2} = K_{y2} = K$). However, the aquifer is heterogeneous in the vertical direction. It has two layers with hydraulic conductivity $K_1 = 3 \times 10^{-6} m/s$ for the upper layer and $K_2 = 2 \times 10^{-4} m/s$ for the lower layer. The specific storage of the aquifer in both regions 1 and 2 is $10^{-4} m^{-1}$. Consider that the pumping well $P_w$ is located at $(609m, 9m)$ in region 1 shown in Figure 2 with a rate of $120 m^3/day$ for one year pumping. The hydraulic head distribution predicted from the MODFLOW simulations is denoted as the dotted line shown in Figure 3. A multi-layered aquifer with heterogeneous hydraulic conductivity may be approximated as an equivalent homogeneous medium. The equivalent hydraulic conductivity $K_h$ may be evaluated as (Schwartz and Zhang, 2003):

$$K_h = \sum_i^m b_i K_i / \sum_i^m b_i \qquad (50)$$

where $K_i$ is the hydraulic conductivity in the horizontal direction for layer $i$, $b_i$ is the thickness of layer $i$, and $m$ is the number of the layers. Accordingly, the equivalent horizontal hydraulic conductivity $K_h$ for the two layered L-shaped aquifer is estimated as $1.2 \times 10^{-4} m/s$. The hydraulic head distribution predicted by the present solution of Eqs. (26) and (27) and represented by the dotted line is shown in Figure 3. The figure indicates that the head distribution simulated by the present solution agrees with that by MODFLOW except in the region near the no-flow boundary FG which has the largest relative deviation 2.1% between these two models. The comparison of the head distributions predicted by the present solution and MODFLOW ensures that the simplification of aquifer layers in the present model is suitable and gives a fairly good predicted results."

- Concerning the model region, the L-shaped domain is surely a big deviation from the real aquifer geometry, especially along boundary AG, but even more along boundaries FE and ED. Thus deviances, as shown in Fig. 3 could be expected. The problem with the manuscript is that it cannot trace back the differences to its causes: it could be the different numerical approach (MODFLOW, FEM, 'analytical') or

the different model region. Were the results of the numerical models obtained with sufficient mesh refinement?

Response:

1. The aquifer geometry in real-world situation could be very complicated. In order to investigate the groundwater flow system in the real-world aquifer, the problem domain is simplified so that the analytical model or numerical model is easy to apply. This study conceptualizes an irregular aquifer in Kihm et al. (2007) as an L-shaped aquifer to simulate the flow due to groundwater pumping by MODFLOW and the present solution. The differences between the finite element solution presented by Kihm et al. (2007) and present solution (or MODFLOW) shown in Figure 3 are significant near the boundaries AG, FE and ED. However, their effects on groundwater head distribution and stream depletion rate near the pumping well are very small because those boundaries are far from the area near the pumping well that we focus on. The discussion on this issue is given in lines 25-36, page 11 as "The head distributions predicted by the FEM solution and present solution have obvious differences in the area far away from the pumping well. Those differences may be mainly caused by the difference in the physical domain considered in FEM solution and the simplified domain made in the present solution. In addition, the mathematical model in Kihm et al. (2007) considered the unsaturated flow and deformation of the unsaturated soil, which may also affect the head distribution after pumping. Notice that the pumping well is very close to the stream boundary AB, which is the main stream in that area and provides a large amount of filtration water to the well. Hence, it seems that the groundwater flows in the region 1 for x ≤ 300m (near boundary AG) and in the region 2 for y ≥ 200m (near boundaries FE and ED) are both far away from the well and almost not influenced by the pumping."

2. Figure C provides the spatial hydraulic head distributions with streamlines after one year pumping simulated by MODFLOW using two different cell sizes, 1m×1m (blue dashed line) and 3m×3m (pink solid line). The result shows no difference while using two different cell sizes, indicating that the cell size 3m×3m used in MODFLOW is good enough to predict the spatial head distribution.

● The production well is located quite near to the boundary AB. It can be expected that the strong head gradients that appear due to this constellation can only be reproduced numerically if strong mesh refinement is used in the direct vicinity of the well.

Response:

We agree that a finer mesh can give better results in the vicinity of the well. We think the mesh size $(3m \times 3\text{m})$ in MODFLOW simulation is relatively small compared to the length of boundary AB (852m) and may give fairly good results. The difference of hydraulic heads near the pumping well predicted by the MODFLOW using cell sizes $(3m \times 3\text{m})$ and $(1m \times 1\text{m})$ is negligibly small as mentioned in previous response. Accordingly, the use of $3m \times 3\text{m}$ mesh in MODFLOW is capable of producing good prediction in head gradients in the area adjacent to the pumping well.

● Concerning the real world situation, it could be doubted that a numerical approach with a constant head boundary can address the physically relevant processes in that case. I would expect that strong or weak connection between aquifer and surface water body play a role in reality in addition.

Response:

We agree that the connection between aquifer and stream has an impact on the groundwater flow in the aquifer, but its impact in reality is strong only in the region near the stream. The Poonggye stream and its tributary are perennial stream and almost fully penetrate the fluvial aquifer system reported in Kihm et al. (2007). Unfortunately there is no information available regarding the streambed properties; thus, we consider that the stream has a prefect hydraulic connection with the aquifer. If the permeability of the streambed is significantly lower than that of the aquifer, then the Robin type condition should be employed as the stream boundary (see, e.g., Huang and Yeh, 2015, 2016). Such a treatment for the stream boundary however is beyond the scope of this study.

● If the paper could be re-written in a way to address the points made, I could deliver a more positive comment.

Response:

Thanks, we have largely revised the manuscript.

References

Huang, C. S., and Yeh, H. D.: Estimating stream filtration from a meandering stream under the Robin condition, Water Resources Research, 51, 4848-4857, doi:10.1002/2015WR016975, 2015.

Huang, C. S., and Yeh, H. D.: An analytical approach for the simulation of flow in a heterogeneous confined aquifer with a parameter zonation structure, Water Resources Research, 52, 9201-9212, doi:10.1002/2016WR019443, 2016.

Kihm, J.-H., Kim, J.-M., Song, S.-H., and Lee, G.-S.: Three-dimensional numerical simulation of fully coupled groundwater flow and land deformation due to groundwater pumping in an unsaturated fluvial aquifer system, Journal of Hydrology, 335, 1-14, http://dx.doi.org/10.1016/j.jhydrol.2006.09.031, 2007.

:

[Figure]

**Figure A: Contours of hydraulic head in L-shaped aquifer predicted by the present solution, MODFLOW, and FEM simulations with irregular outer boundary reported in Kihm et al. (2007).**

[Figure]

**Figure B:** Temporal distributions of hydraulic head $H_{io}$ observed at piezometer $O_i$ and $H_{iF}$ simulated by the FEM simulations both reported in Kihm et al. (2007) and $H_{iA}$ and $H_{iM}$ predicted by the present solution and MODFLOW, respectively, for i = 1 - 3.

[Figure]

**Figure C: Contours of hydraulic head with streamline in L-shaped aquifer simulated by MODFLOW with different cell size.**

---

## Author Comment (AC3) · 9 Jan 2018

D. Ferris david.ferris@usask.ca

The paper under review presents a semi-analytic method for describing groundwater flow in an irregular (L-shaped) unconfined aquifer bounded on two sides by contributing streams. The authors have presented a solution for groundwater flow in a steady-state condition, and, using the steady-state solution as a boundary condition, under the influence of a single pumping well. The authors' work is developed from the work of Kihm et al. in the 2007 paper "Three-dimensional numerical simulation of fully coupled groundwater flow and land deformation due to groundwater pumping in an unsaturated fluvial aquifer system" and draws heavily from the conceptual model developed therein.

Substantive Praise-Worthy Aspects:

1. In this paper, the authors present a novel method for solving for the groundwater flow field for a complex hydrogeology problem. As noted by anonymous referee #1, few papers address groundwater flow in multi-unit aquifers with complex shape, so by presenting a semi-analytic solution to groundwater flow under these conditions, this paper provides insight into methodology for representing hydrologic processes. The problem addressed by the authors also provides insight into modelling the relative contribution of aquifer storage and stream filtration water to the total water abstracted from a pumping well. The authors' work also contributes to an understanding and awareness of the interaction of surface hydrology and groundwater, a topic that should be further addressed and developed. By developing a solution for the groundwater flow field in a system incorporating these factors, the authors have made a worthy contribution to the field of hydrology and engineering.

Response:

Thanks for the comment. We provide a point-by-point response to each of your comments listed below. The page and line numbers mentioned in our responses are referred to those in the revised manuscript.

Substantive Considerations:

2. This paper draws heavily on the work of Kihm et al. (2007), and I am concerned that not all the material presented has been cited correctly. Several examples of incorrectly cited material are provided below:

The sentence on P.4, L.9-10 is cited as a summary, but the wording may not be sufficiently different from the original sentence in Kihm et al. (2007, P.4).

Response:

The sentence in lines 16-17, page 4 has been modified as: "The annual average heads above the bottom of the aquifer are respectively identified as 5.18 m, 4.06 m and 5.29 m at points A, B, and D (Kihm et al., 2007)."

3. A direct quotation from Kihm et al. (2007, P.4) that was not properly indicated or cited was detected on P.4, L.17-18.

Response:

The citation has been added in lines 28-31, page 4 as: "The annual average depth from the ground surface to the water table is 1.26 m with a spatial variation from 0.57 m to 1.95 m in accordance with the average water stages in the streams AB and BD (Kihm et al., 2007). This depth was estimated under the hydrostatic equilibrium condition for the aquifer system before pumping and subject to the effect of net annual average rainfall."

Prior to the start of groundwater pumping, the aquifer system is assumed to be at a hydrostatic equilibrium condition corresponding to the net annual average rainfall rate (i.e., 20%×1287 mm/year = $8.16 \times 10^{-9}$ m/s), the annual average depth to the water table from the ground surface (i.e., 1.26 m), and the annual average water stages above the bottom of the aquifer in the two surrounding perennial streams (i.e., 5.18 m at Point A, 4.06 m at Point B, 5.29 m at Point C), which are all mentioned above.

4. Figure 6 is an updated reprint of Kihm et al.'s Figure 12 (2007, P.12), but is not directly cited in the figure caption.

Response:

Part of Figure 6 is from Kihm et al. (2007) (i.e. observation and FEM simulation at piezometers $O_1$, $O_2$ and $O_3$) and therefore the citation of Kihm et al. (2007) has been added in the figure caption as

"Figure 6: Temporal distributions of hydraulic head $H_{io}$ observed at piezometer $O_i$ and $H_{iF}$ simulated by the FEM simulations both reported in Kihm et al. (2007) and $H_{iA}$ and $H_{iM}$ predicted by the present solution and MODFLOW, respectively, for $i = 1$ - 3."

5. Important assumptions made in the development of the conceptual model have not been discussed. These assumptions follow those made in Kihm et al. (2007) and include the assumption that hydrostatic conditions exist in the vertical profile

through both units of the aquifer (i.e. the piezometric surface is equal to the water table at all points along the vertical profile) and that recharge to the system from vertical percolation or precipitation is negligible. These assumptions, and others, may represent significant deviations from real-world conditions, and should be explicitly stated.

Response:

Thanks for the comment. The pumping well in this study is considered as a fully penetrating well as mentioned in Kihm et al. (2007), and thus the hydraulic gradient in the vertical direction is neglected. Furthermore, the effect of rainfall recharge on the water table had been considered as stated in our response to the third comment. We have modified some sentences in the revised manuscript to make them clear:

In Sect. 2.1:

"Pumping wells in the conceptual model are assumed to fully penetrate the aquifer near the perennial stream AB as those did in Kihm et al. (2007), and therefore the hydraulic gradient in vertical direction is neglected." (lines 24-26, page 4)

In Sect. 2.2:

"Consider that there are totally M pumping wells in region 1 and N pumping wells in region 2, and all the pumping wells fully penetrate the aquifer." (lines 2-3, page 5)

6. Although the piezometer data presented in Kihm et al. 2007 appears to support the modelled solutions, it should be noted that piezometer observations are only available over a period of 5 days; no information is presented to validate the modelled response to pumping beyond this period. Considering Figure 6, the 5-day observation period appears insufficient to observe any response to pumping at piezometer 3 (O3). This indicates that these data are irrelevant for the purpose of validating the transient solution for hydraulic head distribution presented by the authors.

Response:

In general, field observations for groundwater pumping are not easy to obtain and the measurement period is usually limited in a short time (Hunt et al., 2001; Fox, 2004; Lough and Hunt, 2006). We have compared the predicted results of proposed solution to the field data in a period of 5 days and the largest relative difference 1.74% occurs at $O_2$. This result indicates that the present solution gives fairly good predictions in the early pumping period. In addition, the comparison of the temporal head distributions predicted by the present solution

and two different numerical approaches ensures that the present solution also provides reasonably good results in predicting the head distribution after long term pumping. We have rewritten the sentence in lines 8-10, page 12 as:

"In addition, the largest relative differences between measured heads and predicted heads by the present solution at $O_1$ to $O_3$ during 0.1 to 5 day are respectively 1.64%, 1.74% and 0.62%, indicating that the present solution gives good predictions in the early pumping period."

7. It is also to be considered that at a time period of less than 5 days, the majority of the modelled contribution of abstracted water is from aquifer storage (SRR), with the contribution from stream filtration (SDR) increasing after this point. The absence of observed piezometer response to pumping after a time of 5 days would seem to prevent any conclusions from being drawn as to the application of the method presented by the authors in predicting aquifer response to pumping in situations with a large stream filtration component.

Response:

In the short time pumping period (in 5 days), the present solution has been validated by measured data provided by Kihm et al. (2007). Unfortunately, there is no more observation about the pumping response beyond 5 days. The simulation result from FEM for the aquifer system has been verified by Kihm et al. (2007). Figure 6 shows a good match for the predictions of the present solution with the FEM simulations for pumping after 5 days, indicating that the present solution provides a fairly good prediction and is applicable in engineering practice.

8. It is my opinion that the results presented in this paper are insufficient to draw conclusions as to the validity of the methodology presented in predicting aquifer response to pumping. The results presented, however, demonstrate consistency between the semi-analytic method presented by the authors and the numeric model developed by Kihm et al. for the same aquifer system. Likewise, it is my opinion that the results presented are insufficient to draw conclusions regarding the significance of unsaturated flow and land deformation due to the limited observed data.

Response:

The text ", implying that the effects of unsaturated flow and land deformation on the groundwater flow in Yongpoong aquifer are small and may be negligible" in the conclusion has been slightly modified and moved to line 10, page 12 after "in the early pumping period." as

"Moreover, the effects of unsaturated flow and land deformation on the groundwater flow in Yongpoong aquifer are small and may be negligible."

9. The authors present the semi-analytic solution as a design tool for determining well location. The demonstrated applicability of the numeric simulations presented by Kihm et al. 2007 and the authors' solution developed in MODFLOW, validated by the semi-analytic method presented in this paper, would seem to be more flexible and appropriate tools for the design of well location.

Response:

One of the objectives in this study is to interpret the flow interaction between the aquifer and nearby streams, which can be used as a design tool to determine well location with a specific pumping rate for required amounts of SDR from nearby stream. Thus, the calculation of stream depletion rate (SDR) is necessary to determine the well locations based on the estimation of distance to the stream for extracting a specific amount of water from a nearby stream. Basically, the SDR can be easily estimated by taking the derivative of analytical solution with respect to the related direction, then integrating along the stream. However, the numerical approaches presented by Kihm et al. (2007) and MODFLOW are not available to calculate the SDR directly.

Further to the substantive observations which I have made above, there are several additional observations of a less critical nature that I would like to make.

Strengths:

10. The derivation of the analytic solution appears well documented and described. This paper provides the reader with a clear description of the analytical methods used by the authors, theoretically allowing for the results to be reproduced. The literature review presented by the authors also appears to be detailed, and well-structured, providing valuable information to other scientists interested in studying groundwater flow in aquifers with complex boundaries and that are bounded by contributing streams.

Response: Thanks.

Areas of Improvement:

11. The assumption was made that all flow is horizontal, including the flow through the overlying clay loam aquitard unit, which has been assigned a hydraulic conductivity two orders of magnitude lower than the underlying loamy sand unit. This assumption is necessary for the simplification of the groundwater flow

equation to 2-dimensions, but is non-realistic and the implications of this assumption have not been addressed by the authors.

Response:

As described in Kihm et al. (2007) for Yongpoong 2 Agriculture District, the streams almost fully penetrate the aquifer system and a fully penetrating well is installed and screened in the entire aquifer near one of the streams. Accordingly, it is reasonable to treat the flow as horizontal in the aquifer. Furthermore, the equivalent hydraulic conductivity for the loam aquitard and loamy sand units with different conductivities is estimated and used to simulate the flow through these two units.

12. The equivalent hydraulic conductivity for horizontal flow (Eq. (48)), discussed on P.10, L.3-4, is calculated as the weighted arithmetic mean of the two units assuming the full thickness (2.5 m) of the overlying unit is available for groundwater flow. Since the overlying unit is only saturated to a maximum seasonal average thickness of 0.79 m (as described on P.4, L.9), it may be more appropriate to use the saturated thickness of the upper layer when calculating the equivalent hydraulic conductivity for the aquifer.

Response:

Thanks for the comment. We suppose that the thickness of 0.79 m for the overlying unit is a typo and should read 1.79 m (i.e., 5.29 m - 3.5 m=1.79 m). We use 1.79 m for the upper layer thickness to estimate the equivalent hydraulic conductivity ($K_h$). The estimated $K_h$ is $1.3 \times 10^{-4}$. The difference between this figure and the value used in the study (i.e., $1.2 \times 10^{-4}$) is insignificant, implying that the influence of different thickness of overlying unit on the $K_h$ is small.

The logic regarding the required well setback from a stream is incomplete (P.12, L.7-9), and the connection between the required well setback distance from possible contaminants and the well setback distance from a stream is not clear.

Response:

The sentence in lines 4-6, page 13 is rewritten as: "Driscoll (1986, p. 615) mentioned that a well shall be installed at least $45.7 m$ from areas of spray materials, fertilizers or chemicals that contaminate the soil or groundwater. Hence, the distance from the pumping well to the stream is considered at least 50 m to guarantee the quality of extracted water."

13. As noted by anonymous referee #2, the solution presented by the authors is semianalytical. The first use of the term "semi-analytical" by the authors is in the

conclusion on P.12, L.25. The solution presented by the authors should be consistently described throughout the paper, as appropriate.

Response:

Thanks for the suggestion. The steady state solution derived in this study is analytical and the transient solution is semi-analytical because it needs a numerical tool to obtain the time-domain solution. To avoid confusion, we therefore use the word "semi-analytical" in lieu of "transient" before the time-domain solution in the revised manuscript.

Several minor grammatical issues were found within the paper, and are listed as follows:

P.2, L.9 typo: "arbitrarily", should be "arbitrary"

Response: Thanks, it has been corrected.

P.2, L.12-15 ambiguous references; it is not clear that the authors are referring to the work of Kihm et al. (2007)

Response:

We have modified the sentences in lines 14-17, page 2 as: "The domain of the aquifer in their study is in L shape and bounded by streams and impermeable bedrocks. They performed FEM simulations for steady-state spatial distributions of hydraulic head before aquifer pumping and then for the distributions of hydraulic head and land displacement vector after one-year pumping. Their simulation results were compared and validated with the field measurements of hydraulic head and vertical displacement in the transient case."

P.2, L.12 inconsistent hyphenation of "L-shape"

Response: Thanks, it has been corrected.

P.2, L.15 missing "the": "in <the> transient case"

Response: Done as suggested.

P.2, L. 20 poor grammar: "to solve a regional groundwater in an…"

Response:

We have rewritten the sentence in lines 22-24, page 2 as: Serrano (2013) illustrated the use of Adomian's decomposition method to solve a regional groundwater flow problem in an unconfined aquifer bounded by the main stream on one side, two tributaries on two sides, and an impervious boundary on the other side.

P.3, L.31 "perennial stream<s>…"

Response: Corrected.

P.4, L.22 syntax error: "Consider that there are totally M pumping wells…"
(Throughout paper) inconsistent use of italics to denote units, and spaces between values and units (i.e. 6m, 6 m, 6 m)

Response: We have carefully checked and revised the manuscript.

P.2, L.34 ambiguous reference to "irregular boundaries" – what are irregular boundaries?

Response:

    We have modified the sentences in lines 1-3, page 3 as:

"Kuo et al. (1994) applied the image well theory and Theis' equation to estimate transient drawdown in an aquifer with irregularly shaped boundaries. The aquifer is an oil reservoir bounded by three tortuous faults. However, the number of the image wells should be largely increased if the aquifer boundary is asymmetric and rather irregular."

P.3, L.18 ambiguous reference: "principle direction aligned with the border of the sub-region"; which border?

Response: The sentence in lines 24-26, page 3 has been rewritten as: "The aquifer is divided into two rectangular sub-regions. The aquifer in each sub-region is homogeneous but anisotropic in the horizontal plane with principal direction aligned with the borderline of the rectangular sub-regions."

P.13, L.6-7. Awkward transition. This should either be a new paragraph, or these sentences should be rewritten.

Response:

    We have divide the second paragraph in pages 13-14 of conclusion into two parts as:

    "The 3D finite difference model MODFLOW is first used to check the accuracy of hydraulic head predictions by the present solution for the L-shaped two-layered aquifer system. The hydraulic head distributions predicted by present solutions agree fairly well over the entire aquifer except the heads nearing the no-flow boundary. The solution for hydraulic head distribution in the L-shaped aquifer without pumping has been used to investigate the effect of anisotropic ratio ($Kx/Ky$) on the steady-state flow system. It is interesting to

note that the flow pattern in terms of lines of equal hydraulic head is strongly influenced by the value of anisotropic ratio for the region near the turning point of the L-shaped aquifer.

The transient solution proposed by this study is employed to simulate the head distribution induced by pumping in the aquifer within the agriculture area of Gyeonggi-Do, Korea. The aquifer is approximated as L-shaped in this study. The simulation results indicate that the largest relative difference in predicted temporal head distributions at three piezometers by the present solution and Kihm et al.'s (2007) FEM simulation is less than 1.74%, implying that the effects of unsaturated flow and land deformation on the groundwater flow in Yongpoong aquifer are small and may be negligible."

Suggestions:

A careful and detailed review of the entire paper should be conducted by the authors to ensure all material is appropriately cited. The authors should revisit the description of the conceptual model and either further develop and detail the assumptions made in the development of the conceptual model or clearly state that the conceptual model and assumptions have been taken from the work of Kihm et al. (2007) and refer the readers to that paper for details.

The authors should address the implications of the simplifying assumptions when applying the results of the semi-analytic and numeric solutions for groundwater flow in this aquifer to the real world. The limitations of the 5-day observation period should be noted by the authors. The conclusions drawn by the authors should be reconsidered. The results appear to demonstrate consistency between the semi-analytical method presented by the author and the numeric model presented by Kihm et al. (2007), and raise questions as to the significance of unsaturated flow and land deformation. Conclusions regarding real-world aquifer response to groundwater abstraction appear unsupported. It is recommended that the authors reframe their work as a method of validating the numeric simulations and as a method of developing a better understanding of the physical processes governing groundwater flow.

Response:

Thanks for the suggestion. We have revised our work according to the comments herein and the comments from two anonymous referees for manuscript improvement.

Reviewer Experience:

It should be noted that I am a Master of Science candidate in the field of engineering, with minimal experience in either analytical or numerical methods for describing

groundwater flow. I have no prior experience refereeing academic submissions. The observations and opinions I have expressed herein should be considered with my inexperience in mind.

Proposed Fate:

The authors are to be commended for their approach to this complex problem. It is my opinion that the methods and results presented by Lin, Chang, and Yeh makes a valuable contribution to the field of hydrology and engineering and are of interest to the scientific community. However, the issues noted above are significant. I recommend that this paper be resubmitted for review following the revisions suggested above. I would further recommend that extreme caution be exercised by both the authors and by the editor in vetting the submission for incorrectly cited material.

Response: Thanks.

References:

Fox, G. A.: Evaluation of a stream aquifer analysis test using analytical solutions and field data, Journal of the American Water Resources Association, 40(3), 755-763, 10.1111/j.1752-1688.2004.tb04457.x, 2004.

Hunt, B., Weir, J., Clausen, B.: A stream depletion field experiment, Ground Water, 39(2), 283–289, doi/10.1111/j.1745-6584.2001.tb02310.x, 2001.

Lough, H. K., and Hunt, B.: Pumping Test Evaluation of Stream Depletion Parameters, Ground Water, 44(4), 540-546, doi:10.1111/j.1745-6584.2006.00212.x, 2006.

Kihm, J.-H., Kim, J.-M., Song, S.-H., and Lee, G.-S.: Three-dimensional numerical simulation of fully coupled groundwater flow and land deformation due to groundwater pumping in an unsaturated fluvial aquifer system, Journal of Hydrology, 335, 1-14, http://dx.doi.org/10.1016/j.jhydrol.2006.09.031, 2007.

[Figure]

**Figure 6: Temporal distributions of hydraulic head $H_{io}$ observed at piezometer $O_i$ and $H_{iF}$ simulated by the FEM simulations both reported in Kihm et al. (2007) and $H_{iA}$ and $H_{iM}$ predicted by the present solution and MODFLOW, respectively, for $i = 1 - 3$.**

---

## Editor Decision (ED1)

The paper needs an accurate revision, answering some scientific and technical questions.

**Scientific questions**

1) From the discussion in the introduction about the results of Kihm et al. (2007), it is not clear which are the limits of the results given in that paper, which are overcome with the approach proposed in the present paper. In other words, what is wrong with the results of Kihm et al. (2007) and why is it necessary to develop a new model?

2) The results of Figure 6 show that the three tested models (FEM, FDM and analytical solution) provide very similar results; moreover, the gap between models' predictions and the observed data is almost the same for the three models. This is surprising, due to the great differences between the assumptions which are at the basis of the three models. This should be further discussed.

3) At several points of the paper it is claimed that the borders AG and DE are far from the pumping well, so that the flow through these borders does not contribute to the water extracted from the pumping well. On the other hand, it is sometimes stated (e.g., page 9, lines 7 to 10) that "water extracted from the pumping well comes from different sources such as nearby streams, constant-head boundaries, and aquifer storage". Therefore, there is a clear contradiction, which should be corrected.

4) Section 3.5 does not provide particular information, is a quite straightforward application of the proposed procedure and could be erased without compromising the scientific quality of the paper.

**Technical questions**

Page 1, lines 1 to 2. Substitute "in the L-shaped fluvial aquifer" with "in L-shaped fluvial aquifers".

Page 1, lines 19 to 20. Erase the sentence "The SDR solution … pumping rate".

Page 1, line 23. Erase "the" before "groundwater".

Page 2, line 3. Substitute "none of them are to deal" with "none of the cited papers deals".

Page 2, line 5. Substitute "for evaluating the" with "to model".

Page 2, lines 6 to 7. Substitute "heterogeneous aquifer properties" with either "heterogeneous aquifer" or "spatially-variable aquifer properties".

Page 2, line 7. Erase "We therefore… present solution".

Page 2, line 33. SDR has not yet been defined in the text.

Page 3, line 25. Add "assumed to be" before "homogeneous".

Page 3, line 30. Substitute "to inverse Laplace-domain solution" with "to invert the Laplace-domain solution".

Page 3, line 31. Erase "in L-shaped heterogeneous aquifer".

Page 4, line 2. Add ", whose characteristics are " before "reported".

Page 4, lines 3 to 4. Substitute "The west side of the plain is a mountainous area with the formation of exposed impermeable bedrock and the east side has the Poonggye stream which passes the district from the southwest corner toward the northeast", possibly with "The west side of the plain is a mountainous area, where impermeable bedrock outcrops, and the Poonggye stream flows along the east side from the southwest corner toward the northeast corner".

Page 4, line 7. Add "as" before "reported".

Page 4, line 9. Add "s" to "deposit".

Page 4, lines 9 & 10. Substitute "of a thickness" with "with a thickness" (twice).

Page 4, line 13. Add "s" to "coordinate".

Page 4, line 18. Substitute "did in" with "was done by".

Page 4, lines 19 & 20. Substitute "they are not streams and therefore not count for their contribution in the calculations of SDR in Sect. 2.5 Stream depletion rate", possibly with "they do not coincide with streams and therefore do not contribute to SDR as calculated in Sect. 2.5 Stream depletion rate"

Page 4, line 24. Substitute "reported in" with "as reported by".

Page 4, line 27. Rephrase "with suction lift less than 7 m".

Page 4, lines 30 & 31. Rephrase the sentence "This depth… net annual average rainfall".

Page 4, lines 31 & 32. Move the sentence "This aquifer… $\phi_2(x, y, t)$." To the next subsection and possibly rephrase it.

Page 5, lines 2 to 12. Please rephrase these sentences. They can be stated in a more straightforward way.

Page 6, line 4. Is $d_2$ correct in the definition of $Q_{1k}{}^*$?

Page 6, line 6. Add "s" to "region".

Page 9, lines 8 to 13. Substitute "Pumping… the dimensionless solutions", possibly with "Pumping in an aquifer near a stream often produces water filtration from the stream toward the well (Yeh et al., 2008). Water extracted from the pumping well comes from different sources such as nearby streams, constant-head boundaries, and aquifer storage. considered. The extraction rate from the stream is referred to as stream depletion rate (SDR).Since the boundaries AG and ED do not correspond to streams in physical world and are mathematically treated as constant-head because they are far from the pumping well, only the extraction rate from streams AB and BD near the pumping well needs to be considered. The dimensionless solutions".

Page 9, lines 13, 14 and following. $\widetilde{SDR}_A$ and $\widetilde{SDR}_B$ should be replaced with $\widetilde{SDR}_{AB}$ and $\widetilde{SDR}_{BD}$.

Page 9, line 22. SRR has not yet been defined in the text.

Page 10, line 3. Add "(USGS, 2005)" after "MODFLOW".

Page 10, lines 4 to 6. Erase "The MODFLOW… (USGS, 20015)."

Page 10, line 10. Substitute "The upper loam layer is 2.5 m and lower sand layer 3.5 m" with "The upper loam layer is 2.5-meter-thick and the lower sand layer is 3.5-meter-thick".

Page 10, lines 11 & 12. Substitute "The types of outer boundary" with "The boundary conditions".

Page 10, lines 13 & 14. Substitute "The fluvial aquifer reported in Kihm et al. (2007) is isotropic and homogeneous in horizontal direction." With "Following Kihm et al. (2007), the fluvial aquifer is considered isotropic and homogeneous in the horizontal direction."

Page 10, lines 15 &16. There is some confusion among $K$, $K_1$ and $K_2$. The notation should be changed.

Page 10, lines 19 & 20, 27. Modify the sentences "The hydraulic head distribution… in Figure 3" and "represented by the dotted line is shown in Figure 3". Check the correspondence of the line types in the map and what is written in the text. Moreover, it should be clearly stated that contour lines are drawn in Figure 3.

Page 10, lines 21 & 22. Substitute "A multi-layered aquifer… (Charbeneau, 2000):", possibly with "The global behavior of a multi-layered aquifer may be approximated with that of an equivalent homogeneous medium, whose hydraulic conductivity in the horizontal plane $K_h$ may be evaluated as (Charbeneau, 2000):"

Page 10, lines 27 & 28. Rephrase "The figure indicates… except the region".

Page 10, line 30 & 31. Rephrase "MODFLOW ensures… predicted results."

Page 11, line 18. Erase "Note that".

Page 11, line 28. Substitute "made" with "used".

Page 11, lines 31 to 33. Rephrase "Hence… by the pumping".

Page 12, line 2. Add "as" before "mentioned".

Page 12, line 3. Should "upper" be substituted with "northern"?

Page 12, line 5. Add "simulations" after "MODFLOW".

Page 12, line 11. Rephrase "decline greater".

Figure 4. Why are not MODFLOW results shown?

---

## Author Response (AR2)

**Reply to Editor's comment**

The paper needs an accurate revision, answering some scientific and technical questions.
Scientific questions
Response:

Thanks, we provide point-by-point response to each of your comment listed below. The page and line numbers given in our responses are referred to those in the revised manuscript.

A. From the discussion in the introduction about the results of Kihm et al. (2007), it is not clear which are the limits of the results given in that paper, which are overcome with the approach proposed in the present paper. In other words, what is wrong with the results of Kihm et al. (2007) and why is it necessary to develop a new model?

Response:

Thanks for the suggestion. Kihm et al. (2007) developed a complicate mathematical model with considering unsaturated flow and solid skeleton deformation. Their model was approximated based on the Galerkin FEM to a set of four coupled nonlinear equations in terms of one head variable and three displacement variables. Thus, their numerical solution requires a large amount of hydrogeologic information as input data and computer time in solving the simultaneous nonlinear equations. The present analytical solution, on the other hand, delivers fairly good results even the aquifer layers are approximated as one layer and both the effects of unsaturated flow and solid skeleton deformation on the groundwater flow are neglected. We add following sentences in lines 14-17, page 2 to illustrate the limitations of the work of Kihm et al. (2007): "Their simulation results were compared and validated with the field measurements of hydraulic head and vertical displacement in the transient case. Note that their model was approximated based on the Galerkin FEM to a set of four coupled nonlinear equations in terms of one head variable and three displacement variables. Thus, their numerical solution requires a large amount of hydrogeologic information as input data and computer time in solving the simultaneous nonlinear equation."

B. The results of Figure 6 show that the three tested models (FEM, FDM and analytical solution) provide very similar results; moreover, the gap between models' predictions and the observed data is almost the same for the three models. This is surprising, due to the great differences between the assumptions which are at the basis of the three models. This should be further discussed.

Response:

The close results obtained from these three models can be attributed two facts. The first is because the average thickness of unsaturated zone is only 1.26 m, which may not exert a large impact on the saturated flow system. The second is due to fact that the pumping well is very close to the main stream (stream AB) of the fluvial aquifer and the extracted water mainly comes from the stream. The pumping drawdowns in three piezometers are small in comparison to the aquifer thickness, implying that the influence of land displacement may be neglected. Thus, the temporal head distributions of three models (FEM, MODFLOW and analytical model) have similar patterns and deliver close predictions. We have added some sentences in line 33, page 11 and lines 1-3, page 12 to discuss this issue: "This figure indicates that the hydraulic heads predicted by the present solution has a good agreement with those simulated by Kihm et al. (2007). This is due to the fact that the average thickness of unsaturated zone is about 1.26 m, which may not affect the saturated flow system. Moreover, the pumping well is very close to the main stream (stream AB) of the fluvial aquifer. The extracted water mainly comes from the stream. The largest pumping drawdown is 0.66m at $O_1$, implying that the influence of land displacement may be neglected."

C. At several points of the paper it is claimed that the borders AG and DE are far from the pumping well, so that the flow through these borders does not contribute to the water extracted from the pumping well. On the other hand, it is sometimes stated (e.g., page 9, lines 7 to 10) that "water extracted from the pumping well comes from different sources such as nearby streams, constant-head boundaries, and aquifer storage". Therefore, there is a clear contradiction, which should be corrected.

Response:

Thanks for the comment. To avoid confusion, the phrase " … such as nearby streams, constant-head boundaries, and aquifer storage" in line 9, page 9 is changed to: " … such as aquifer storage and nearby streams"

D. Section 3.5 does not provide particular information, is a quite straightforward application of the proposed procedure and could be erased without compromising the scientific quality of the paper.

Response:

This section and associated descriptions in the text and figure are all removed.

Technical questions

The questions (problems) with numbers 1-7, 9-12, 14-20, 26, 28, 30-34, 37, 40, 41, 43, and 45 listed below have been all corrected as suggested.

1. Page 1, lines 1 to 2. Substitute "in the L-shaped fluvial aquifer" with "in L-shaped fluvial aquifers".

2. Page 1, lines 19 to 20. Erase the sentence "The SDR solution … pumping rate".

3. Page 1, line 23. Erase "the" before "groundwater".

4. Page 2, line 3. Substitute "none of them are to deal" with "none of the cited papers deals".

5. Page 2, line 5. Substitute "for evaluating the" with "to model".

6. Page 2, lines 6 to 7. Substitute "heterogeneous aquifer properties" with either "heterogeneous aquifer" or spatially-variable aquifer properties".

7. Page 2, line 7. Erase "We therefore… present solution".

9. Page 3, line 25. Add "assumed to be" before "homogeneous".

10. Page 3, line 30. Substitute "to inverse Laplace-domain solution" with "to invert the Laplace-domain solution".

11. Page 3, line 31. Erase "in L-shaped heterogeneous aquifer".

12. Page 4, line 2. Add ", whose characteristics are " before "reported"

14. Page 4, line 7. Add "as" before "reported".

15. Page 4, line 9. Add "s" to "deposit".

16. Page 4, lines 9 & 10. Substitute "of a thickness" with "with a thickness" (twice).

17. Page 4, line 13. Add "s" to "coordinate".

18. Page 4, line 18. Substitute "did in" with "was done by".

19. Page 4, lines 19 & 20. Substitute "they are not streams and therefore not count for their contribution in the calculations of SDR in Sect. 2.5 Stream depletion rate", possibly with "they do not coincide with streams and therefore do not contribute to SDR as calculated in Sect. 2.5 Stream depletion rate"

20. Page 4, line 24. Substitute "reported in" with "as reported by".

26. Page 6, line 6. Add "s" to "region".

28. Page 9, lines 13, 14 and following. $\widetilde{SDR}_A$ and $\widetilde{SDR}_B$ should be replaced with $\widetilde{SDR}_{AB}$ and $\widetilde{SDR}_{BD}$.

30. Page 10, line 3. Add "(USGS, 2005)" after "MODFLOW".

31. Page 10, lines 4 to 6. Erase "The MODFLOW… (USGS, 20015)."

32. Page 10, line 10. Substitute "The upper loam layer is 2.5 m and lower sand layer 3.5 m" with "The upper loam layer is 2.5-meter-thick and the lower sand layer is 3.5-meter-thick".

33. Page 10, lines 11 & 12. Substitute "The types of outer boundary" with "The boundary conditions".

34. Page 10, lines 13 & 14. Substitute "The fluvial aquifer reported in Kihm et al. (2007) is isotropic and homogeneous in horizontal direction." With "Following Kihm et al. (2007), the fluvial aquifer is considered isotropic and homogeneous in the

horizontal direction."

37. Page 10, lines 21 & 22. Substitute "A multi-layered aquifer… (Charbeneau, 2000):", possibly with "The global behavior of a multi-layered aquifer may be approximated with that of an equivalent homogeneous medium, whose hydraulic conductivity in the horizontal plane $K_h$ may be evaluated as (Charbeneau, 2000):"

40. Page 11, line 18. Erase "Note that".

41. Page 11, line 28. Substitute "made" with "used".

43. Page 12, line 2. Add "as" before "mentioned".

45. Page 12, line 5. Add "simulations" after "MODFLOW".

8.  Page 2, line 33. SDR has not yet been defined in the text.

Response:

  The phrase "stream depletion rate" has been added before the acronym *SDR*. (line 32, page2)

13. Page 4, lines 3 to 4. Substitute "The west side of the plain is a mountainous area with the formation of exposed impermeable bedrock and the east side has the Poonggye stream which passes the district from the southwest corner toward the northeast", possibly with "The west side of the plain is a mountainous area, where impermeable bedrock outcrops, and the Poonggye stream flows along the east side from the southwest corner toward the northeast corner".

Response:

  The sentence after the phrase "in Kihm et al. (2007)." in lines 3-4, page 4 has been rewritten as suggested.

21. Page 4, line 27. Rephrase "with suction lift less than 7 m".

Response:

  We have modified the sentence as: "Todd and Mays (2005, p. 232) noticed that the discharge rates in a shallow well may range up to $500 \, \text{m}^3/\text{day}$ $(0.01 \, \text{m}^3/\text{s})$ and the suction lifts should be less than 7 m for efficient and continuous service." (lines 26-28, page 4)

22. Page 4, lines 30 & 31. Rephrase the sentence "This depth… net annual average rainfall".

Response:

  It is rephrased as: "The depth for the aquifer system before pumping was estimated under the hydrostatic equilibrium condition and with considering the effect of net annual average rainfall." (lines 30-32, page 4)

23. Page 4, lines 31 & 32. Move the sentence "This aquifer…$\phi_2(x, y, t)$." To the next subsection and possibly rephrase it.

Response:

The sentence is moved to the beginning of Sect. 2.2 and modified as: "As shown in Figure 2, the aquifer is divided into two sub-regions named as regions 1 and 2 and variables $\phi_1(x, y, t)$ and $\phi_2(x, y, t)$ are their corresponding hydraulic heads. Consider that there … penetrate the aquifer." (lines 2-3, page 5)

24. Page 5, lines 2 to 12. Please rephrase these sentences. They can be stated in a more straightforward way.

Response:

These sentences have been modified as (lines 3-13, page 5):

"… Consider that there are totally $M$ pumping wells in region 1 and $N$ pumping wells in region 2, and all the pumping wells fully penetrate the aquifer. The coordinates of $k$th well in region 1 and $l$th well in region 2 are denoted as $(x_{1k}, y_{1k})$ and $(x_{2l}, y_{2l})$, respectively, and the pumping rate per unit thickness at $k$th well is represented by $Q_{1k}$ $[L^2/T]$ and that at $l$th well is $Q_{2l}$ $[L^2/T]$. The governing equations describing 2D hydraulic head distributions in region 1 and region 2 are respectively expressed as:

$$K_{x1}\frac{\partial^2 \phi_1}{\partial x^2} + K_{y1}\frac{\partial^2 \phi_1}{\partial y^2} = S_{s1}\frac{\partial \phi_1}{\partial t} - \sum_{k=1}^{M} Q_{1k}\delta(x - x_{1k})\delta(y - y_{1k})$$

$$\leq x \leq l_1, 0 \leq y \leq d_1 \tag{1}$$

$$K_{x2}\frac{\partial^2 \phi_2}{\partial x^2} + K_{y2}\frac{\partial^2 \phi_2}{\partial y^2} = S_{s2}\frac{\partial \phi_2}{\partial t} - \sum_{l=1}^{N} Q_{2l}\delta(x - x_{2l})\delta(y - y_{2l})$$

$$l_2 \leq x \leq l_1, d_1 \leq y \leq d_2 \tag{2}$$

where $K_x[L/T]$ and $K_y[L/T]$ are respectively the hydraulic conductivities in $x$- and $y$-direction, and $S_s[L^{-1}]$ is the specific storage. The symbol $\delta$ represents one dimensional (1D) Dirac's delta function $[1/T]$."

25. Page 6, line 4. Is $d_2$ correct in the definition of $Q_{1K}$?

Response:

Yes, it is correct.

27. Page 9, lines 8 to 13. Substitute "Pumping… the dimensionless solutions", possibly with "Pumping in an aquifer near a stream often produces water filtration from the stream toward the well (Yeh et al., 2008). Water extracted from the pumping well comes from different sources such as nearby streams, constant-head boundaries, and aquifer storage. considered. The extraction rate from the stream is referred to as stream depletion rate (SDR). Since the boundaries AG and ED do not correspond to streams in physical world and are mathematically treated as constant-head because they are far from the pumping well, only the extraction rate from streams AB and BD near the pumping well needs to be considered. The dimensionless solutions."

Response:

Thanks for the suggestion. The sentences have been rephrased in lines 8-12, page 9 as: "… Water extracted d by the pumping well comes from the sources such as aquifer storage and nearby streams. The extraction rate from the stream is referred to as stream depletion rate (*SDR*). Since the boundaries AG and ED do not correspond to streams in physical world and are mathematically treated as constant-head because they are far from the pumping well, only the water filtration from streams AB and BD to the nearby pumping well needs to be considered."

29. Page 9, line 22. SRR has not yet been defined in the text.

Response:

The phrase "storage release rate" has been added before the acronym *SRR*. (line 21, page 9)

35. Page 10, lines 15 &16. There is some confusion among K, K1 and K2. The notation should be changed.

Response:

To avoid confusion, the hydraulic conductivities for upper and lower layers are respectively changed to $K_U$ and $K_L$ and its associated sentence is changed to: "It has two layers with hydraulic conductivity $K_U = 3 \times 10^{-6} \text{ m/s}$ for the upper layer and $K_L = 2 \times 10^{-4} \text{ m/s}$ for the lower layer." (line 11-12, page 10)

36. Page 10, lines 19 & 20, 27. Modify the sentences "The hydraulic head distribution… in Figure 3" and "represented by the dotted line is shown in Figure 3". Check the correspondence of the line types in the map and what is written in the text. Moreover, it should be clearly stated that contour lines are drawn in Figure 3.

Response:

The sentence in lines 13-14, page 10 is modified as: "Figure 3 shows the hydraulic head distribution obtained from MODFLOW simulations and denoted as the dotted line." Moreover, the sentence in lines 20-21, page 10 is rephrased as: "The solid line in Figure 3 represents the hydraulic head distribution predicted by the present solution of Eqs. (26) and (27)."

38. Page 10, lines 27 & 28. Rephrase "The figure indicates… except the region".

Response:

The sentence in lines 21-22, page 10 is rewritten as: "The head distribution predicted by the present solution agrees with that of MODFLOW simulations except in the region near the no-flow boundary FG, where has the largest relative deviation 2.1%."

39. Page 10, line 30 & 31. Rephrase "MODFLOW ensures… predicted results."

Response:

The sentence in lines 22-24, page 10 is rephrased as: "The comparison of the head

distributions indicates that the use of equivalent hydraulic conductivity in the present model is appropriate and gives a fairly good predicted results."

42. Page 11, lines 31 to 33. Rephrase "Hence… by the pumping"

Response:

The sentences are modified as: "Notice that the pumping well … water to the well. Hence, both groundwater flows in region 1 for $x \leq 300$ m (near boundary AG) and in region 2 for $y \geq 200$ m (near boundaries DE and EF) are almost not influenced by the pumping because these two regions are far away from the pumping well." (lines 25-27, page 11)

44. Page 12, line 3. Should "upper" be substituted with "northern"?

Response:

No, the boundary FG is actually at the west of the fluvial aquifer as shown in Figure 1. To avoid confusion, we add a north arrow in Figure 2 (also shown below) and modify the sentence as: "Note that $O_1$ was installed near the stream AB while $O_3$ was far away from the stream but close to the impermeable boundary FG." (lines 29-30, page 11)

46. Page 12, line 11. Rephrase "decline greater"

Response:

It has been rephrased as: "The hydraulic head at $O_1$ drops larger than those at $O_2$ and $O_3$ whereas the former is located closer to the pumping well $P_w$." (line 9, page 12)

47. Figure 4. Why are not MODFLOW results shown?

Response:

Thanks for the suggestion. Figure 4 is redrawn (also shown below) to add the steady-state hydraulic head contours simulated by the MODFLOW. The associated text in lines 27-33, page 10 is revised as: "Kihm et al. (2007) reported … 
[revised manuscript text omitted]

---

## Author Response (AR3)

Comments to the Author:

The paper has been revised, but unfortunately, the answers to the posed questions are not fully satisfactory.

Response:

 We provide point-by-point response to the comments listed below. The page and line numbers mentioned in our responses are referred to those in the revised manuscript.

In particular, I recall the following two basic questions.

(1) Page 2, lines 11 to 17. This paragraph should be rewritten, because the sentences which have been added are not very clear and do not provide a very logical argument.

Response:

 We revise the text to explain what Kihm et al. (2007) did and point out the defect in their solution. The text has been rewritten in lines 10-17, page 2 as: "The domain of the aquifer is in L shape and bounded by streams and impermeable bedrock. Their mathematical model was developed with considering the unsaturated flow and solid skeleton deformation to a set of four coupled nonlinear equations in terms of one head variable and three displacement variables. The Galerkin FEM was employed to simulate the steady-state spatial distribution of hydraulic head before aquifer pumping and then the distributions of hydraulic head and land displacement vector after one-year pumping. Their simulation results were compared and validated with the field measurements of hydraulic head and vertical displacement in the transient case. However, it was a lot of work to solve the simultaneous nonlinear equation and might difficult be applied because it took a lot of calculation time and resources for long-term simulations."

 Furthermore, we have added some sentences in lines 22-28, page 3 to explain the reason why a new model is necessary and illustrate the merit of proposed study: "The impacts of groundwater extraction by wells should therefore be thoroughly investigated before pumping. Theoretically, numerical methods, such as FEM are good tools to simulate groundwater flow in irregular-shaped aquifer such as the approximately L-shaped aquifer in Kihm et al. (2007). However, they are generally time consuming and demanding of computing power (Younger, 2007), and it is necessary to re-generate the mesh of problem domain for different size of L-shaped aquifers. The analytical solution can be easily applied for different size of L-shaped aquifer with similar boundaries and properties by replacing the length or width of the solution. Thus, an analytical solution is proposed in this study to evaluate the spatiotemporal distribution of the drawdown at any location in the L-shaped aquifer.

This paper develops a 2D mathematical model for describing the groundwater flow in an approximately L-shaped fluvial aquifer which is very close to the case of numerical simulations reported in Kihm et al. (2007)."

(2) Page 11, line 33 to page 12, line 3. These statements are not supported by a rigorous demonstration.

Response:

We have added more statements to discuss the comparison of results of FEM and present solution in lines 11-29, page 12 as: "This figure indicates that the hydraulic heads predicted by the present solution has a good agreement with those simulated by Kihm et al. (2007). The largest relative differences between the temporal hydraulic head predicted by the FEM and present solution are 0.58%, 0.31%, and 0.51% m for $O_1$, $O_2$, and $O_3$, respectively. As shown in Figures 2 and 3, more than half of aquifer boundary is surrounded by perennial streams and constant-head boundaries, and the pumping well is installed close to the main stream AB for stream filtration and screened in the highly permeable sand layer (Kihm, et al. 2007). It is expected that the extracted water from the pumping well mainly comes from the stream AB and the differences of the hydraulic heads from the observations ($O_1$, $O_2$, and $O_3$) between the present solution and FEM simulation are insignificant because the boundary geometry near these observation in both present solution and Kihm et al. (2007) are similar. These difference may be caused by the solid skeleton deformation and unsaturated flow considered in the FEM simulation of Kihm et al. (2007). The Young's modulus ($E$) of the sand layer is $1.9 \times 10^7$ N/m$^2$ (Kihm et al. 2007), implying that the deformation of sand layer due to pumping may be small. This is also responded in the vertical displacements reported in Kihm et al. (2007). The largest vertical displacement reported in Kihm et al. (2007) is only -0.003 m. Hence, the effect of land displacement during pumping may not significantly influence the hydraulic heads in piezometers. In addition, the unsaturated flow may slightly affect the saturated flow system. This is because the average thickness of unsaturated zone is about 1.26 m, and it consists of a low permeability material (i.e., loam), which is two order less than that of saturated zone (i.e., most of sand). As mentioned earlier, the pumping well is screened in the sand layer and the stream AB is the major source of extracted water. Therefore, the influence of unsaturated zone on the saturated flow may be small. Under such hydrogeological conditions, the present solution yields similar prediction for the temporal hydraulic head distribution as compared with those of FEM. Compared with the field observation, the differences of predicted hydraulic head among FEM, present solution and MODFLOW are all less than 0.08 m at

these three piezometers during 0.1 to 10 day." Moreover, we have added some sentences in lines 8-12, page 14, to discuss the comparison of temporal hydraulic heads predicted by the present solution and FEM as: "The transient solution for head distribution is employed to simulate the head distribution induced by pumping in the aquifer within the agriculture area of Gyeonggi-Do, Korea. The aquifer is approximated as L-shaped in this study. The present solution delivers fairly good result in predicting the temporal hydraulic head distribution while comparing with those of FEM reported in previous study. Those simulation results seem to indicate that the effects of unsaturated flow and land displacement on the groundwater flow are not significant and may be ignorable. The largest relative differences between the measured heads and the predicted heads by the present solution at three piezometers are less than 1.74%."

A specific questions about equation (48): is tilde missing above each quantity?
Response:

Thanks for the comment. To avoid confusion, we add following sentences in lines 22-24, page 9 as: "Further, the *SDR* solution for streams AB and BD in real time domain are respectively denoted as $SDR_{AB}$ and $SDR_{BD}$ and also obtained by the Stehfest algorithm (Stehfest, 1970). The total dimensionless stream depletion rate ($SDR_T$) in time domain comes from the streams (AB and BD) is expressed as:

$$SDR_T = SDR_{AB} + SDR_{BD} \tag{48}"$$

I hope that the authors can fix these problems, so that the paper will be finally acceptable for publication.
Non-public comments to the Author:
In the previous round of reviewing, I mentioned that "a deep revision of the whole text is required", well behind the technical comments that were explicitly listed. However, only those comments have been considered and therefore a complete and accurate revision of the text is still missing. Therefore, I recall you that such a complete and accurate text revision, together with appropriate answers to the questions mentioned in the public comments, is mandatory to allow me to finally accept the paper for publication. Otherwise, I will be very sad, but I would be obliged to take a negative decision.
Response:

We have addressed all of the questions mentioned above. In addition, the primary comments/suggestions by three reviewers and editor along with the text regarding to the comments in the revised manuscript are listed and given at the end of this reply.

References:

[revised manuscript text omitted]

| real aquifer geometry, especially along boundary AG, but even more along boundaries FE and DE. | | distribution after pumping. Notice that the pumping well is very close to the stream boundary AB, which is the main stream in that area and provides a large amount of filtration water to the well. Hence, both groundwater flows in region 1 for $x \leq 300$ m (near boundary AG) and in region 2 for $y \geq 200$ m (near boundaries DE and EF) are almost not influenced by the pumping because these two regions are far away from the pumping well." |
|---|---|---|
| **Comments and suggestions of Short comment#1** | Lines and pages in the revised manuscript | The text regarding to the comment/suggestion in the revised manuscript. |
| 1. The hydrostatic conditions exist in the vertical profile through both units of the aquifer (i.e. the piezometric surface is equal to the water table at all points along the vertical profile) and that recharge to the system from vertical percolation or | 1. lines 29-31, page 4

2. lines 9-10, page 5 | 1. "Pumping wells in the conceptual model are assumed to fully penetrate the aquifer near the perennial stream AB as mentioned in Kihm et al. (2007), and therefore the hydraulic gradient in vertical direction is neglected."
2. "Consider that there are totally $M$ pumping wells in region 1 and $N$ pumping wells in region 2, and all the pumping wells fully penetrate the aquifer." |

| precipitation is negligible. | | |
|---|---|---|
| 2. No information is presented to validate the modelled response to pumping beyond period of 5 days. | lines 30-33, page 12 | "In addition, the largest relative differences between measured heads and predicted heads by the present solution at $O_1$ to $O_3$ during 0.1 to 5 day are respectively 1.64%, 1.74% and 0.62%, indicating that the present solution gives good predictions in the early pumping period." |
| 3. The results presented are insufficient to draw conclusions regarding the significance of unsaturated flow and land deformation due to the limited observed data. | lines 11-33, page 12 to line 1, page 13 | "This figure indicates that the hydraulic heads predicted by the present solution has a good agreement with those simulated by Kihm et al. (2007). The largest relative differences between the temporal hydraulic head predicted by the FEM and present solution are 0.58%, 0.31%, and 0.51% m for $O_1$, $O_2$, and $O_3$, respectively. As shown in Figures 2 and 3, more than half of aquifer boundary is surrounded by perennial streams and constant-head boundaries, and the pumping well is installed close to the main stream AB for stream filtration and screened in the highly permeable sand layer (Kihm, et al. 2007). It is expected that the extracted water from the pumping well mainly comes from the stream AB and the differences of the hydraulic heads from the observations ($O_1$, $O_2$, and $O_3$) between the present solution and FEM simulation are insignificant because the boundary geometry near these observation in both present solution and Kihm et al. (2007) are similar. These difference may be caused by the solid skeleton deformation and unsaturated flow considered in the FEM simulation of Kihm et al. (2007). The Young's modulus ($E$) of the sand layer is $1.9 \times 10^7$ N/m$^2$ (Kihm et al. 2007), implying that the deformation of sand layer due to pumping may be small. This is also responded in the vertical displacements |

| | | |
|---|---|---|
| | | reported in Kihm et al. (2007). The largest vertical displacement reported in Kihm et al. (2007) is only -0.003 m. Hence, the effect of land displacement during pumping may not significantly influence the hydraulic heads in piezometers. In addition, the unsaturated flow may slightly affect the saturated flow system. This is because the average thickness of unsaturated zone is about 1.26 m, and it consists of a low permeability material (i.e., loam), which is two order less than that of saturated zone (i.e., most of sand). As mentioned earlier, the pumping well is screened in the sand layer and the stream AB is the major source of extracted water. Therefore, the influence of unsaturated zone on the saturated flow may be small. Under such hydrogeological conditions, the present solution yields similar prediction for the temporal hydraulic head distribution as compared with those of FEM. Compared with the field observation, the differences of predicted hydraulic head among FEM, present solution and MODFLOW are all less than 0.08 m at these three piezometers during 0.1 to 10 day. In addition, the largest relative differences between measured heads and predicted heads by the present solution at $O_1$ to $O_3$ during 0.1 to 5 day are respectively 1.64%, 1.74% and 0.62%, indicating that the present solution gives good predictions in the early pumping period. Moreover, the effects of unsaturated flow and land deformation on the groundwater flow in Yongpoong aquifer are small and may be neglected." |
| **Comments and suggestions of Editor on 02/08 and 0310** | Lines and pages in the revised manuscript | The text regarding to the comment/suggestion in the revised manuscript. |
| 1. What is wrong with the results of | 1. lines 8-17, page 2 2. lines 22-28, page | 1. "Kihm et al. (2007) used a general multidimensional hydrogeomechanical Galerkin FEM to analyze three-dimensional (3D) problems of saturated-unsaturated flow and land displacement |

| | | |
|---|---|---|
| Kihm et al. (2007) and why is it necessary to develop a new model. | 3 | induced by pumping in a fluvial aquifer in Yongpoong 2 Agriculture District, Gyeonggi-Do, Korea. The domain of the aquifer is in L shape and bounded by streams and impermeable bedrock. Their mathematical model was developed with considering the unsaturated flow and solid skeleton deformation to a set of four coupled nonlinear equations in terms of one head variable and three displacement variables. The Galerkin FEM was employed to simulate the steady-state spatial distribution of hydraulic head before aquifer pumping and then the distributions of hydraulic head and land displacement vector after one-year pumping. Their simulation results were compared and validated with the field measurements of hydraulic head and vertical displacement in the transient case. However, it was a lot of work to solve the simultaneous nonlinear equation and might difficult be applied because it took a lot of calculation time and resources for long-term simulations." 2. The reason why a new model is necessary is described as follows: "The impacts of groundwater extraction by wells should therefore be thoroughly investigated before pumping. Theoretically, numerical methods, such as FEM are good tools to simulate groundwater flow in irregular-shaped aquifer such as the approximately L-shaped aquifer in Kihm et al. (2007). However, they are generally time consuming and demanding of computing power (Younger, 2007), and it is necessary to re-generate the mesh of problem domain for different size of L-shaped aquifers. The analytical solution can be easily applied for different size of L-shaped aquifer with similar boundaries and properties by replacing the length or width of the solution. Thus, an analytical solution is proposed in this study to evaluate the spatiotemporal distribution of the drawdown at any location in the L-shaped aquifer. This paper develops a 2D mathematical model … in Kihm et al. (2007)." |
| 2. The results of | 1. lines 11-29, page | 1. "This figure indicates that the hydraulic heads predicted by the present solution has a good |

[revised manuscript text omitted]